# NaRA: Noise-Aware LoRA for Parameter-Efficient Fine-Tuning of Diffusion LLMs

**Shuaidi Wang**[1]  **Zhan Zhuang**[1,2]  **Ruping Huang**[1,3]  **Yu Zhang**[1]

## Abstract

Diffusion Large Language Models (dLLMs) have emerged as a promising non-autoregressive generative paradigm. Given the prohibitive computational cost of full fine-tuning, Parameter-Efficient Fine-Tuning (PEFT) has become the standard approach. However, existing PEFT methods (*e.g.*, LoRA), originally tailored for autoregressive models, rely on static parameters that are agnostic to the noise level. Consequently, they ignore the intrinsic dynamics of the diffusion process, where input distributions and generation difficulty shift significantly along the denoising trajectory, rendering them suboptimal for dLLMs. To address this, we propose **N**oise-**a**ware Low-**R**ank **A**daptation (NaRA), which introduces a low-rank core matrix generated by a lightweight, globally shared hypernetwork conditioned on the noise level. This design enables the update matrices to vary continuously along the diffusion process while keeping parameter and latency overhead negligible. We provide a theoretical justification for the proposed NaRA framework and empirically demonstrate consistent improvements over noise-agnostic baselines across commonsense reasoning, mathematical reasoning, and code generation benchmarks. Our code is available at https://github.com/generaldi/NaRA.

## 1. Introduction

Large language models (LLMs) have transformed natural language processing, predominantly relying on the autore-

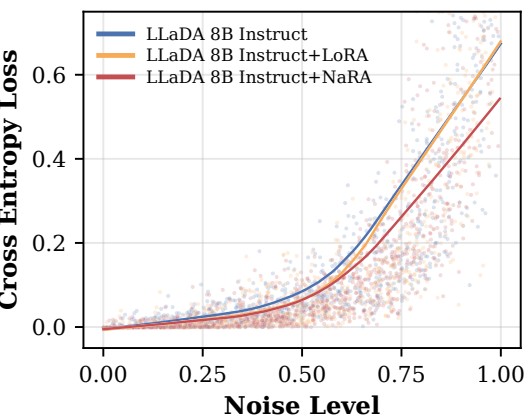

*Figure 1.* The cross-entropy loss of LLaDA (Nie et al., 2025) across noise levels. Scatter points denote per-example losses and solid curves show smoothed trends estimated via Locally Weighted Scatterplot Smoothing (LOWESS). LoRA yields most of its gain at mid-noise levels, whereas NaRA consistently reduces loss across a broader range of noise levels by adapting to denoising dynamics.

gressive (AR) paradigm for sequential, token-by-token generation (OpenAI, 2022; Achiam et al., 2023; Yang et al., 2024; 2025). Despite their competitive performance, AR models are constrained by their inherent sequential nature, which limits inference parallelism and restricts the attention mechanism to unidirectional context. Diffusion large language models (dLLMs) (DeepMind, 2025; Inception Labs et al., 2025; Nie et al., 2025; Zhu et al., 2025a;b; Ye et al., 2025) have emerged as an alternative to address these limitations. Formulated as masked diffusion models (Austin et al., 2021a; Lou et al., 2024), dLLMs diverge from the AR paradigm by employing a multi-step denoising process. By iteratively refining a masked sequence with bidirectional attention (Vaswani et al., 2017), dLLMs support parallel token generation and global planning over the whole context (Zhao et al., 2025), while offering a tunable computation-quality trade-off through the denoising process (Israel et al., 2025).

Fully fine-tuning these large pre-trained dLLMs for downstream tasks remains computationally prohibitive (Hu et al., 2022; Dettmers et al., 2023). A closely related limitation has been observed in autoregressive LLMs, where Parameter-Efficient Fine-Tuning (PEFT) has emerged as the dominant paradigm for adaptation. PEFT strategies (Lester et al.,

---

[1]Department of Computer Science and Engineering, Southern University of Science and Technology, Shenzhen, China [2]Department of Computer Science, City University of Hong Kong, Hong Kong, China [3]Department of Computer Science and Engineering, Hong Kong University of Science and Technology, Hong Kong, China. Correspondence to: Yu Zhang <yu.zhang.ust@gmail.com>.

*Proceedings of the 43rd International Conference on Machine Learning*, Seoul, South Korea. PMLR 306, 2026. Copyright 2026 by the author(s).

2021; Liu et al., 2022; 2024; Chen et al., 2025; Huang et al., 2025a) adapt pre-trained models by updating only a small set of additional parameters, thereby bypassing the computational and storage bottlenecks of full fine-tuning. Among them, Low-Rank Adaptation (LoRA) (Hu et al., 2022) has gained widespread adoption in LLMs due to its strong empirical performance and simplicity. Motivated by these successes, a natural strategy is to naively extend such efficient techniques to dLLMs.

Despite its efficiency, standard LoRA is structurally mismatched to dLLMs because it imposes a *noise-agnostic* low-rank update throughout the entire denoising process. This is suboptimal, as the inherently noise-dependent nature of diffusion models is ignored (Balaji et al., 2022; Choi et al., 2022). We empirically substantiate this effect on LLaDA (Nie et al., 2025) for mathematical reasoning tasks, with experimental details provided in Appendix L. As illustrated in Figure 1, while standard LoRA effectively reduces cross-entropy loss in low-to-medium noise regimes, it yields negligible improvement at high noise levels when compared to the baseline. A primary reason for this phenomenon is that standard LoRA employs a fixed set of adapter weights throughout the entire denoising process. This approach neglects the significant shifts in prediction scope and modeling difficulty that occur in diffusion models. Specifically, standard LoRA was originally designed for autoregressive models where the objective remains consistent, namely predicting the probability of the next single token. In contrast, dLLMs require modeling the probabilities of a varying number of target tokens throughout the denoising process. Furthermore, since the number of mask tokens decreases progressively throughout the denoising process, the semantic information within the input evolves significantly across the denoising trajectory, leading to considerable variations in reconstruction difficulty. Consequently, the observed failure of static adapters to generalize across all noise levels underscores a fundamental limitation: a one-size-fits-all update is insufficient for dLLMs. In light of these factors, designing adaptive update weights tailored to the specific demands of each noise level represents a more theoretically grounded and effective strategy for dLLMs.

A natural way to achieve the noise-awareness for LoRA is to train distinct LoRA adapters for different timesteps (Zhuang et al., 2025). However, this is parameter-inefficient and ignores shared structure across noise levels. We empirically support this with a Multi-LoRA baseline. This baseline splits the noise range into four intervals. Even though it uses about four times more parameters than standard LoRA, it still underperforms standard LoRA, as discussed in Section 6.5. To bypass these issues, we propose **N**oise-**a**ware **Lo**w-**R**ank **A**daptation (NaRA). Inspired by Zhuang et al. (2024), NaRA replaces the fixed LoRA adapter with a dynamic structure that inserts a noise-aware core matrix be-

tween two static projection matrices. To generate the core matrix, a lightweight, globally shared hypernetwork is used based on the noise level. Consequently, the weight updates in NaRA vary continuously across different noise levels. This design effectively addresses the aforementioned challenges. By modeling the core matrix as a continuous function, NaRA naturally captures dependencies across noise levels through the diffusion process in dLLMs without training disjoint adapters.

We theoretically establish that multiple distinct update matrices can be equivalently represented within our proposed decomposed framework. Nevertheless, we empirically validate its efficacy through extensive experiments on the LLaDA family, utilizing both Base and Instruct models. Our evaluation spans three distinct domains, including commonsense reasoning, mathematical reasoning, and code generation. The results demonstrate that NaRA consistently achieves state-of-the-art performance across diverse benchmarks while maintaining negligible computational overhead. Furthermore, our analysis confirms that the hypernetwork effectively modulates the update magnitude in correlation with the noise level, validating the necessity of noise-aware adaptation in dLLMs.

Our main contributions are summarized as follows.

- (*Problem Identification*) To the best of our knowledge, we are the first to identify the limitation of applying standard PEFT methods, originally designed for autoregressive models, to dLLMs. We point out that these standard PEFT methods fail to account for the inherently noise-dependent nature of the diffusion process.

- (*The NaRA Framework*) We propose NaRA, the first noise-aware PEFT framework tailored for dLLMs. By utilizing a lightweight hypernetwork to generate a core matrix based on the noise level, NaRA captures noise-dependent dynamics with negligible parameter overhead.

- (*Theoretical and Empirical Validation*) We provide a theoretical analysis demonstrating NaRA's expressive power. Furthermore, extensive experiments and ablation studies across multiple benchmarks show that NaRA consistently outperforms standard LoRA and other noise-agnostic PEFT baselines.

## 2. Related work

**Diffusion Models for Language Generation.** Recent dLLMs, including LLaDA (Nie et al., 2025), have reached competitive performance against autoregressive LLMs (DeepMind, 2025; Inception Labs et al., 2025; Song et al., 2025; Ye et al., 2025; Zhu et al., 2025b). A growing body of research has investigated post-training strategies to

unlock specific capabilities, including reinforcement learning for reasoning (Huang et al., 2025b; Pan et al., 2025; Tang et al., 2025; Wang et al., 2025; Zhu et al., 2025a; Zhao et al., 2025; Zekri & Boullé, 2025) and supervised fine-tuning for domain adaptation (Xie et al., 2025; Gong et al., 2025). While these frameworks utilize PEFT to reduce computational costs, they predominantly apply standard LoRA without modification, thereby treating the adaptation process as static. However, this static treatment overlooks the intrinsic dynamics of masked diffusion, which is what the proposed NaRA method aims to address.

**Dynamic Adaptation in PEFT.** For autoregressive models, static PEFT methods have proven highly effective by learning a constant adaptation weight (Hu et al., 2022; Liu et al., 2024; Meng et al., 2024; Huang et al., 2025a; Wang et al., 2026). However, this static assumption yields suboptimal results in diffusion models (Zhang et al., 2023; Ganjdanesh et al., 2024; Zhuang et al., 2024; Ma et al., 2025; Zhuang et al., 2025), where the input distribution dynamically shifts across the denoising trajectory. While Zhuang et al. (2025) has attempted to address this by partitioning timesteps into intervals and employing LoRA-based Mixture-of-Experts (MoE), such approaches introduce significant training difficulties and structural complexity.

# 3. Preliminaries

## 3.1. Masked Diffusion LLM

Consider a training instance $\mathbf{x}_0 = (\mathbf{p}, \mathbf{r}_0)$, where the condition prompt $\mathbf{p}$ and the target response $\mathbf{r}_0$ are sequences of discrete tokens. The forward process gradually corrupts the response $\mathbf{r}_0$ into noise by replacing tokens with a special symbol [MASK]. Formally, at an arbitrary timestep $t \in (0, 1]$, we sample a corrupted state $\mathbf{r}_t$ by independently masking each token in $\mathbf{r}_0$ with probability $t$. The transition probability is defined as

$$q(r_t^{(i)} \mid r_0^{(i)}) = \begin{cases} t & \text{if } r_t^{(i)} = \text{[MASK]}, \\ 1 - t & \text{if } r_t^{(i)} = r_0^{(i)}, \end{cases} \quad (1)$$

where $r_t^{(i)}$ denotes the token at the $i$-th position in $\mathbf{r}_t$. Given that the condition prompt remains uncorrupted, the effective noise level is determined solely by the target response. We formally define the noise level $\lambda = m/L_s \in [0, 1]$, where $m$ is the count of masked tokens and $L_s$ denotes the length of the response segment.

Conversely, the generative reverse process employs a neural network with parameters $\theta$ to estimate the conditional probability $p_\theta(r_0^{(i)} \mid \mathbf{p}, \mathbf{r}_t)$ of the original tokens at these masked positions. To achieve this, the model is trained using

a re-weighted cross-entropy loss over the masked tokens

$$\mathcal{L}(\theta) = \mathbb{E}_{t,\mathbf{p},\mathbf{r}_0,\mathbf{r}_t} \left[ \frac{1}{t} \sum_{i \in \mathcal{M}_t} -\log p_\theta(r_0^{(i)} \mid \mathbf{p}, \mathbf{r}_t) \right], \quad (2)$$

where $\mathcal{M}$ indicates the set of masked indices in $\mathbf{r}_t$. Theoretically, Eq. (2) constitutes a variational upper bound on the negative log-likelihood of the data (Shi et al., 2024; Ou et al., 2024).

## 3.2. Low-Rank Adaptation (LoRA)

LoRA (Hu et al., 2022) is a parameter-efficient fine-tuning technique based on the hypothesis that weight updates in pre-trained models reside in a low-dimensional subspace. Consider a weight matrix $\mathbf{W}_0 \in \mathbb{R}^{d \times k}$. LoRA parametrizes the weight update $\Delta\mathbf{W}$ by decomposing it into the product of two low-rank matrices $\mathbf{B} \in \mathbb{R}^{d \times r}$ and $\mathbf{A} \in \mathbb{R}^{r \times k}$, where the rank $r \ll \min(d, k)$. The modified forward pass maps an input $\mathbf{x} \in \mathbb{R}^k$ to the layer output $\mathbf{h} \in \mathbb{R}^d$ as

$$\mathbf{h} = \mathbf{W}_0\mathbf{x} + \Delta\mathbf{W}\mathbf{x} = \mathbf{W}_0\mathbf{x} + \mathbf{B}\mathbf{A}\mathbf{x}. \quad (3)$$

# 4. Methodology

In this section, we present **NaRA**, a noise-aware PEFT method for dLLMs. We detail its dynamic low-rank formulation, a lightweight hypernetwork that enables adaptation across noise levels, theoretical analysis, training and inference details.

## 4.1. Noise-Aware Low-Rank Adaptation

We propose NaRA to incorporate dynamic adaptation capabilities directly into the low-rank bottleneck. Inspired by Zhuang et al. (2024), rather than training separate adapters for distinct noise level intervals, we employ a single dynamic core matrix $\mathbf{C}(\lambda) \in \mathbb{R}^{r \times r}$ positioned between two static update matrices, where $\lambda$ is the noise level. Specifically, for an input $\mathbf{x}$ processed at a specific $\lambda$, NaRA computes

$$\mathbf{h} = \mathbf{W}_0\mathbf{x} + \Delta\mathbf{W}(\lambda)\mathbf{x} = \mathbf{W}_0\mathbf{x} + \mathbf{B}\mathbf{C}(\lambda)\mathbf{A}\mathbf{x}. \quad (4)$$

In this formulation, the matrices $\mathbf{A} \in \mathbb{R}^{r \times k}$ and $\mathbf{B} \in \mathbb{R}^{d \times r}$ remain static, serving the same role as in standard LoRA. However, unlike standard LoRA, our approach introduces $\mathbf{C}(\lambda)$ to explicitly model the dependency of update matrices $\Delta\mathbf{W}(\lambda)$ on the noise level. Specifically, through the matrix multiplication $\mathbf{B}\mathbf{C}(\lambda)\mathbf{A}$, the noise-dependent core matrix $\mathbf{C}(\lambda)$ effectively controls the magnitude and direction of the entire update matrix $\Delta\mathbf{W}(\lambda)$, enabling NaRA to dynamically adjust its behavior according to $\lambda$.

Although update matrices $\mathbf{A}$ and $\mathbf{B}$ are shared across different noise levels, the incorporation of $\mathbf{C}(\lambda)$ enables NaRA

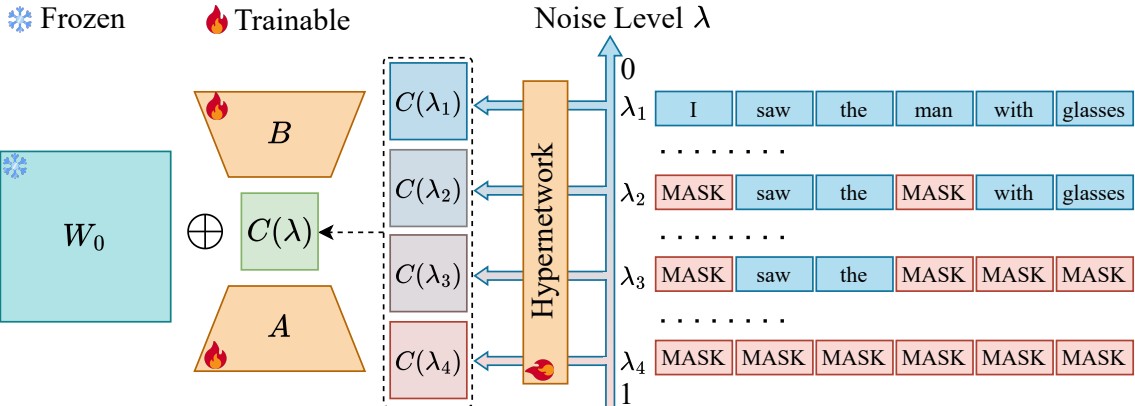

*Figure 2.* Illustration of the architecture in NaRA. The framework dynamically modulates weight updates according to the noise level $\lambda$. Given the noise level $\lambda$ at each denoising step during generation, a hypernetwork generates the dynamic core matrix $\mathbf{C}(\lambda)$, which is then integrated into the low-rank structure, sandwiched between static projection matrices $\mathbf{B}$ and $\mathbf{A}$.

to achieve an expressive power comparable to training independent LoRAs. The following theorem formalizes this equivalence, guaranteeing that NaRA can represent a set of LoRA update matrices. We defer the full proof to Appendix A.

**Theorem 4.1.** *Given a set of $N$ arbitrary weight update matrices $\mathcal{W} = \{\Delta\mathbf{W}_1, \ldots, \Delta\mathbf{W}_N\}$ with each $\Delta\mathbf{W}_i \in \mathbb{R}^{d \times k}$, suppose $r$ satisfies $r \geq \max(\dim(\bigcup_{i=1}^N \mathcal{C}(\Delta\mathbf{W}_i)), \dim(\bigcup_{i=1}^N \mathcal{R}(\Delta\mathbf{W}_i)))$, where $\mathcal{C}(\cdot)$ and $\mathcal{R}(\cdot)$ denote the column and row spaces, respectively, $\bigcup$ denotes the union operation, and $\dim(\cdot)$ denotes the dimension of a linear subspace. Then, there exist shared matrices $\mathbf{B} \in \mathbb{R}^{d \times r}$ and $\mathbf{A} \in \mathbb{R}^{r \times k}$, and a sequence of core matrices $\{\mathbf{C}_1, \ldots, \mathbf{C}_N\}$ with $\mathbf{C}_i \in \mathbb{R}^{r \times r}$, such that*

$$\Delta\mathbf{W}_i = \mathbf{B}\mathbf{C}_i\mathbf{A}, \quad \forall i \in \{1, \ldots, N\}. \tag{5}$$

Although Theorem 4.1 theoretically requires the rank $r$ to be no less than the dimension of the union of the target subspaces, we posit that the effective dimension of this union is relatively low. This is because the weight updates across different noise levels are intrinsically correlated rather than orthogonal, as they address the same generation task. Consequently, there is a significant overlap between their row and column spaces.

Beyond the shared subspace argument above, we further provide a theoretical intuition for why noise-level-specific parameterization is particularly essential for dLLMs compared to AR models. Let $\mathcal{U} = \{1, \ldots, L\}$ denote all token positions. At noise level $\lambda$, a subset $\mathcal{S} \subseteq \mathcal{U}$ of size $m = \lfloor L(1-\lambda) \rfloor$ serves as the revealed positions, and masked positions $\mathcal{M} = \mathcal{U} \setminus \mathcal{S}$ are to be predicted. For AR models, $\mathcal{S}$ is always the prefix $\{1, \ldots, m\}$, yielding a unique input configuration per noise level. For dLLMs, however, $\mathcal{S}$ can be any of $\binom{L}{m}$ subsets, making the num-

ber of possible input configurations exponentially larger. This combinatorial explosion means dLLMs face a far more diverse input distribution at each noise level than AR models, making noise-level-specific parameterization essential rather than merely beneficial.

## 4.2. Hypernetwork-based Noise Integration

As illustrated in Figure 2, the architecture of NaRA relies on a globally shared hypernetwork (Ha et al., 2016) to learn the mapping from the continuous noise level $\lambda$ to the update matrices. To mitigate the spectral bias inherent in low-dimensional coordinates (Rahaman et al., 2019), we first map the noise level $\lambda$ into a $d$-dimensional space using Gaussian Fourier embeddings (Tancik et al., 2020). Specifically, let $\mathbf{k} \in \mathbb{R}^{d/2}$ be a fixed vector sampled from a Gaussian distribution $\mathcal{N}(0, \sigma^2)$. The embedding $\mathbf{e}_\lambda$ is defined as

$$\mathbf{e}_\lambda = \cos(2\pi\mathbf{k}\lambda) \oplus \sin(2\pi\mathbf{k}\lambda), \tag{6}$$

where $\oplus$ denotes vector concatenation. This embedding serves as the input to a lightweight hypernetwork denoted by $\mathcal{F}_\phi$, parameterized by $\phi$, which is implemented as a Multi-Layer Perceptron (MLP). To ensure the training stability, we formulate the core matrix $\mathbf{C}(\lambda)$ as additive deviation from the identity matrix:

$$\mathbf{C}(\lambda) = \mathbf{I}_r + \eta \cdot \mathcal{F}_\phi(\mathbf{e}_\lambda), \tag{7}$$

where $\mathbf{I}_r \in \mathbb{R}^{r \times r}$ denotes the $r \times r$ identity matrix, $\mathcal{F}_\phi$ is to project the embedding $\mathbf{e}_\lambda$ into the latent matrix space $\mathbb{R}^{r \times r}$, and $\eta$ is a scalar hyperparameter that scales the magnitude of the additive deviation.

A key feature of NaRA is the global sharing of the hypernetwork across all layers. Specifically, considering a specific target module $m$ at layer index $l$ within the dLLM, which

was omitted for brevity in Eq. (4), the corresponding weight update $\Delta \mathbf{W}_{l,m}(\lambda)$ is derived as

$$\Delta \mathbf{W}_{l,m}(\lambda) = \mathbf{B}_{l,m} \mathbf{C}(\lambda) \mathbf{A}_{l,m}. \tag{8}$$

Note that while $\mathbf{A}_{l,m}$ and $\mathbf{B}_{l,m}$ vary by layer and module to capture local adaptation patterns, the hypernetwork $\mathcal{F}_\phi$ (and thus $\mathbf{C}(\lambda)$) is shared across the entire model. This sharing strategy offers two critical advantages. First, it ensures efficiency in both the number of parameters and inference latency. By computing $\mathbf{C}(\lambda)$ only once per timestep and broadcasting it to all modules, we ensure that the computational cost remains comparable to standard LoRA. Second, this sharing strategy serves as a regularizer that stabilizes the training and prevents overfitting. Specifically, in contrast to allocating unique hypernetworks to separate layers, which would drastically inflate the optimization landscape, the shared design enforces a unified and robust noise-aware adaptation mechanism, as validated in Section 6.3. Furthermore, the noise-conditioning mechanism introduced here is not restricted to NaRA's low-rank structure. It can be seamlessly applied to other low-rank PEFT methods. As a demonstration, we integrate it with DoRA (Liu et al., 2024), with results reported in Appendix M.

### 4.3. Gradient Analysis and Initialization

To guarantee optimization stability, we compare the gradient dynamics of NaRA with standard LoRA. For standard LoRA, derived from Eq. (3), the gradients with respect to $\mathbf{A}$ and $\mathbf{B}$ are calculated as

$$\nabla_{\mathbf{A}}\mathcal{L} = \mathbf{B}^\top \nabla_{\Delta \mathbf{W}}\mathcal{L}, \quad \nabla_{\mathbf{B}}\mathcal{L} = \nabla_{\Delta \mathbf{W}}\mathcal{L}\mathbf{A}^\top, \tag{9}$$

where $\nabla$ denotes the gradient operator, $\mathcal{L}$ denotes the loss function defined in Eq. (2) and the superscript $\top$ represents the matrix transpose operation. In contrast, NaRA incorporates $\mathbf{C}(\lambda)$ as defined in Eq. (4), resulting in the following gradients as

$$\begin{aligned} \nabla_{\mathbf{A}}\mathcal{L} &= \mathbf{C}(\lambda)^\top \mathbf{B}^\top \nabla_{\Delta \mathbf{W}}\mathcal{L}, \\ \nabla_{\mathbf{B}}\mathcal{L} &= \nabla_{\Delta \mathbf{W}}\mathcal{L}\mathbf{A}^\top \mathbf{C}(\lambda)^\top. \end{aligned} \tag{10}$$

By comparing Eq. (9) and Eq. (10), it is evident that if $\mathbf{C}(\lambda)$ acts as an identity matrix, the gradient of NaRA becomes mathematically identical to that of standard LoRA. Motivated by this, we aim to initialize the hypernetwork such that $\mathbf{C}(\lambda) = \mathbf{I}$ at the start of training. According to Eq. (7), this condition is satisfied when the output of the hypernetwork $\mathcal{F}_\phi(\mathbf{e}_\lambda)$ is zero for any $\lambda$. To achieve this, we initialize the weights and biases of the final layer of the hypernetwork to zero. For other layers, we employ Kaiming uniform initialization (He et al., 2015). Furthermore, consistent with the standard LoRA, the update matrix $\mathbf{A}$ follows the Kaiming initialization, whereas $\mathbf{B}$ is initialized to zero.

### 4.4. Training and Inference

**Training.** NaRA seamlessly integrates with the standard supervised fine-tuning paradigm by introducing a lightweight operation before the forward pass, which involves dynamically generating and broadcasting the modulation matrix $\mathbf{C}(\lambda)$. This design preserves the original optimization pipeline, rendering NaRA a plug-and-play module. The complete training procedure is detailed in Appendix B.

**Inference.** We follow the semi-autoregressive (semi-AR) sampling strategy from LLaDA (Nie et al., 2025), where the sequence is generated iteratively in blocks. To further reduce the latency, we introduce a block-wise early termination mechanism. Specifically, during the decoding, if an end-of-sentence (EOS) token is detected within a block, the model generates at most one additional block and then stops, padding the remaining positions with EOS tokens. This strategy could eliminate redundant computations for non-informative tokens, significantly accelerating the inference while maintaining the generation accuracy. A quantitative analysis of this speedup is provided in Appendix C. Note that this block-wise early termination is a general inference optimization applicable to any adapter-based method including standard LoRA.

## 5. Experiments

In this section, we empirically evaluate the proposed NaRA.

### 5.1. Experimental Setups

**Models and Datasets.** We evaluate two variants of LLaDA (Nie et al., 2025): LLaDA-8B-Instruct, instruction-tuned on 4.5 million instruction–response pairs, and LLaDA-8B-Base, a pretraining-only model without instruction tuning that serves as a more challenging backbone for assessing fine-tuning effectiveness. We conduct evaluations on three domains: commonsense reasoning, mathematical reasoning, and code generation.

For commonsense reasoning, we fine-tune on Commonsense170k (Hu et al., 2023), comprising 170,420 query-answer pairs. We evaluate on eight benchmarks with diverse capabilities, including BoolQ (Clark et al., 2019), PIQA (Bisk et al., 2020), SIQA (Sap et al., 2019), HellaSwag (Zellers et al., 2019), Winogrande (Sakaguchi et al., 2021), OpenBookQA (Mihaylov et al., 2018), ARC-Challenge, and ARC-Easy (Clark et al., 2018). For mathematical reasoning, we fine-tune on Math14k (Hu et al., 2023), a composite dataset derived from GSM8K (Cobbe et al., 2021) and AQuA (Ling et al., 2017) enriched with GPT-generated rationales. We evaluate on four tasks, namely GSM8K, AddSub (Hosseini et al., 2014), AQuA (Ling et al., 2017), and MultiArith (Roy & Roth, 2016). For code generation, we train on the filtered Code-

*Table 1.* Results on commonsense reasoning benchmarks. We report the zero-shot accuracy for all tasks. For fair comparison, all adapter-based methods share the same rank $r = 32$. The "Param" column indicates the percentage of trainable parameters relative to the base model. For each column, the best result is highlighted in **bold**, and the second-best result is underlined.

| METHOD | PARAM(%) | ARC-C | ARC-E | BOOLQ | HELLA | OBQA | PIQA | SIQA | WINO | **AVG** |
|---|---|---|---|---|---|---|---|---|---|---|
| | | | | LLaDA-8B-INSTRUCT | | | | | | |
| ZERO-SHOT | / | 66.55 | 71.00 | 55.60 | 74.41 | 53.60 | 82.75 | 41.56 | 0.08 | 55.69 |
| PROMPT TUNING | 0.001 | 71.16 | 78.58 | 54.31 | 72.44 | 62.20 | 82.48 | 53.12 | 0.24 | 59.32 |
| P-TUNING | 0.417 | 83.70 | 94.53 | 66.18 | 89.21 | 85.80 | 84.11 | 77.38 | 78.69 | 82.45 |
| LoRA | 0.417 | 75.74 | 87.84 | **68.02** | 90.52 | 77.87 | 85.40 | 75.38 | 80.69 | 80.18 |
| HiRA | 0.417 | **86.12** | 95.03 | 66.14 | 87.09 | 85.73 | **85.69** | 78.22 | 77.85 | 82.73 |
| NaRA (OURS) | 0.425 | 85.92 | **95.65** | 67.31 | **90.89** | **86.87** | 85.18 | **79.82** | **81.06** | **84.09** |
| | | | | LLaDA-8B-BASE | | | | | | |
| ZERO-SHOT | / | 55.72 | 65.45 | 59.57 | 55.05 | 42.40 | 66.59 | 33.16 | 46.72 | 53.08 |
| PROMPT TUNING | 0.001 | 56.06 | 66.08 | 50.52 | 40.53 | 47.00 | 57.40 | 37.87 | 52.17 | 50.95 |
| P-TUNING | 0.417 | 83.62 | 92.97 | 57.61 | 88.64 | 85.00 | 85.20 | 75.95 | **80.03** | 81.13 |
| LoRA | 0.417 | 76.99 | 87.30 | **67.17** | 75.15 | 73.33 | **85.53** | 68.76 | 74.59 | 76.10 |
| HiRA | 0.417 | 85.58 | 95.02 | 63.43 | 89.70 | 84.33 | 84.48 | 77.58 | 77.01 | 82.14 |
| NaRA (OURS) | 0.425 | **86.29** | **95.50** | 66.35 | **90.98** | **87.13** | 85.42 | **79.38** | 79.58 | **83.83** |

*Table 2.* Experimental results on mathematical reasoning tasks. We report the zero-shot accuracy for all tasks. For fair comparison, all adapter-based methods share the same rank $r = 32$. The "Param" column indicates the percentage of trainable parameters relative to the base model. **Bold** indicates the best result, and underline indicates the second best.

| METHOD | PARAM(%) | GSM8K | ADDSUB | AQUA | MULTIARITH | **AVG** |
|---|---|---|---|---|---|---|
| | | | LLaDA-8B-INSTRUCT | | | |
| ZERO-SHOT | / | 75.68 | 87.12 | 50.20 | 97.50 | 77.63 |
| PROMPT TUNING | 0.001 | 65.96 | 57.22 | 50.39 | 45.83 | 54.85 |
| P-TUNING | 0.417 | 77.41 | **89.87** | 54.33 | 96.94 | 79.64 |
| LoRA | 0.417 | 78.96 | 88.38 | 54.45 | 98.50 | 80.07 |
| HiRA | 0.417 | 75.01 | 89.62 | 53.94 | 93.72 | 78.07 |
| NaRA (OURS) | 0.425 | **79.03** | 88.53 | **57.27** | **98.72** | **80.89** |
| | | | LLaDA-8B-BASE | | | |
| ZERO-SHOT | / | 37.76 | 32.91 | 35.43 | 37.33 | 35.86 |
| PROMPT TUNING | 0.001 | 20.92 | 15.70 | 30.31 | 26.00 | 23.23 |
| P-TUNING | 0.417 | 73.09 | **87.09** | **49.74** | 93.83 | 75.94 |
| LoRA | 0.417 | 75.28 | 84.73 | 49.61 | 95.56 | 76.29 |
| HiRA | 0.417 | 68.61 | 82.28 | 40.55 | 87.17 | 69.65 |
| NaRA (OURS) | 0.425 | **77.69** | 82.95 | **49.74** | **97.72** | **77.02** |

Feedback dataset (Meng et al., 2024; Zheng et al., 2024) and evaluate on the HumanEval (Chen, 2021) and MBPP (Austin et al., 2021b) benchmarks. Detailed descriptions and statistics of those datasets are provided in Appendix K.

**Baselines and Evaluation.** We compare NaRA against Prompt Tuning (Lester et al., 2021), P-Tuning (Liu et al., 2022), LoRA (Hu et al., 2022), and HiRA (Huang et al., 2025a). For both commonsense and mathematical reasoning tasks, we use the accuracy as the evaluation metric. For code generation, we specifically utilize the lm-evaluation-harness framework (Gao et al., 2024) to assess performance using the pass@1 metric.

**Implementation Details.** We fine-tune models for 10 epochs on mathematical reasoning datasets, while training for 1 epoch on the larger code and commonsense datasets. A

validation set of 128 samples is reserved from each dataset to monitor the validation loss every 64 update steps, allowing us to select the best-performing checkpoint and prevent overfitting. For statistical reliability, we conduct three independent runs for NaRA and report the average performance. To ensure fair comparison, we control the number of trainable parameters to be comparable to that of baselines. Unless otherwise specified, the scaling factor $\eta$ in Eq. (7) is set to 0.1. Additional training and inference details are provided in Appendix G and Appendix H.

### 5.2. Results on Commonsense Reasoning

Table 1 summarizes the accuracy across eight commonsense reasoning benchmarks. On the LLaDA-Instruct backbone, NaRA achieves a state-of-the-art average accuracy of 84.09%, significantly outperforming the standard LoRA

*Table 3.* Pass@1 accuracy on code generation tasks using LLaDA-8B-Instruct. All adapter-based methods use rank $r = 32$. **Bold** indicates the best result, and underline indicates the second best.

| METHOD | MBPP | HUMANEVAL | AVG |
|---|---|---|---|
| ZERO-SHOT | 36.40 | 36.59 | 36.49 |
| PROMPT TUNING | 37.00 | 18.29 | 27.65 |
| P-TUNING | 34.27 | 36.79 | 35.53 |
| LORA | 37.20 | 39.43 | 38.32 |
| HIRA | 27.00 | 39.02 | 33.01 |
| NARA (OURS) | **38.60** | **40.04** | **39.32** |

*Table 4.* Ablation on the scaling factor $\eta$ evaluated on mathematical reasoning tasks. We report the zero-shot accuracy for all tasks.

| $\eta$ | GSM8K | ADDSUB | AQUA | MULTIARITH | AVG |
|---|---|---|---|---|---|
| 1.0 | **79.13** | 87.27 | 56.49 | 98.67 | 80.39 |
| 0.1 | 79.03 | 88.53 | **57.27** | **98.72** | **80.89** |
| 0.01 | 78.85 | **89.63** | 54.46 | 98.11 | 80.26 |

baseline (80.18%) and surpassing advanced variants like HiRA (82.73%).

We further extend the evaluation to the LLaDA-Base backbone to assess NaRA's ability. While LoRA drops to an average of 76.10%, NaRA maintains high performance with an average of 83.83%, and outperforms HiRA (82.14%). In this setting, NaRA ranks the first in 5 out of 8 tasks and second in the remaining 3 tasks, showcasing consistent adaptability. Notably, compared to standard LoRA, NaRA introduces less than 0.01% additional trainable parameters relative to the base model, confirming the efficiency of our noise-aware adaptation strategy.

### 5.3. Results on Mathematical Reasoning

Table 2 summarizes the performance on mathematical reasoning benchmarks. On the LLaDA-Instruct backbone, NaRA achieves the highest average accuracy of 80.89%, surpassing all baseline methods. Notably, NaRA demonstrates superior reasoning capabilities. Crucially, it secures the top position on the challenging AQuA dataset (57.27%), significantly outperforming the second-best method (54.45%), and also leads on GSM8K and MultiArith.

This advantage extends to the LLaDA-Base model, where NaRA maintains the leading position with a top average accuracy of 77.02%. In this setting, NaRA again ranks the first in 3 out of 4 datasets, including GSM8K, MultiArith, and AQuA. These results validate that our noise-aware adaptation effectively enhances mathematical reasoning capabilities compared to standard PEFT methods.

To rule out the possibility that NaRA's improvement is attributable to its marginally larger parameter budget, we additionally evaluate LoRA at rank $r = 40$, which exceeds NaRA's budget at $r = 32$. As shown in Appendix F, LoRA (r=40) still underperforms NaRA (r=32), confirming that the gains stem from noise-aware adaptation rather than parameter scale.

### 5.4. Results on Code Generation

We further evaluate the code generation capabilities on the MBPP and HumanEval benchmarks using the LLaDA-

Instruct backbone. As shown in Table 3, NaRA achieves the best performance with an average pass@1 accuracy of 39.32%. Notably, on the HumanEval dataset, NaRA is the only method that exceeds 40%, outperforming LoRA (39.94%) and other methods. Similarly, on MBPP, NaRA also ranks the first with 38.60% accuracy. These results show that our noise-aware dynamic adaptation effectively captures the structural dependencies required for coding tasks, consistently outperforming existing PEFT baselines.

### 5.5. Impact of Noise Level on Update Magnitude

We analyze the learned update matrices $\Delta W(\lambda)$ defined in Eq. (4) to characterize NaRA's adaptation behavior across noise levels. Since $\Delta W(\lambda)$ is directly added to the pre-trained model parameters, the Frobenius norm of $\Delta W(\lambda)$, denoted $\|\Delta W(\lambda)\|_F$, provides a principled measure of the update magnitude induced by NaRA. Consequently, correlating $\|\Delta W(\lambda)\|_F$ with the noise level $\lambda$ offers a direct lens into how $\lambda$ influences the update matrices. Figure 3 presents the update magnitudes across attention modules for the code generation task, revealing a consistent positive correlation with $\lambda$. Under low-noise conditions, NaRA applies only minor parameter updates through the update matrices $\Delta W(\lambda)$, thereby preserving pre-trained knowledge and reducing the risk of overfitting. As the noise level increases, the magnitude of these updates grows accordingly, providing the model with sufficient capacity to recover semantic structure from heavily masked inputs. Overall, these indicate that NaRA learns an effective noise-aware adaptation strategy.

## 6. Ablation Study

### 6.1. Impact of the Scaling Factor $\eta$

We investigate the sensitivity of NaRA to the scaling factor $\eta$ introduced in Eq. (7). Beyond regulating the magnitude of the core matrix $\mathbf{C}(\lambda)$, $\eta$ serves as a linear scaling coefficient for the gradient flow backpropagated to the hypernetwork $\mathcal{F}_\phi$. Specifically, based on Eq. (10), the gradient of the loss $\mathcal{L}$ with respect to the parameters $\phi$ is formulated as

$$\nabla_\phi \mathcal{L} = \eta \cdot (\mathbf{B}^\top \nabla_{\Delta \mathbf{W}} \mathcal{L} \mathbf{A}^\top) \frac{\partial \mathcal{F}_\phi}{\partial \phi}, \qquad (11)$$

indicating that $\eta$ effectively controls the gradient flow.

Table 4 presents the results on mathematical reasoning tasks. It's worth noting that all three NaRA configurations achieve

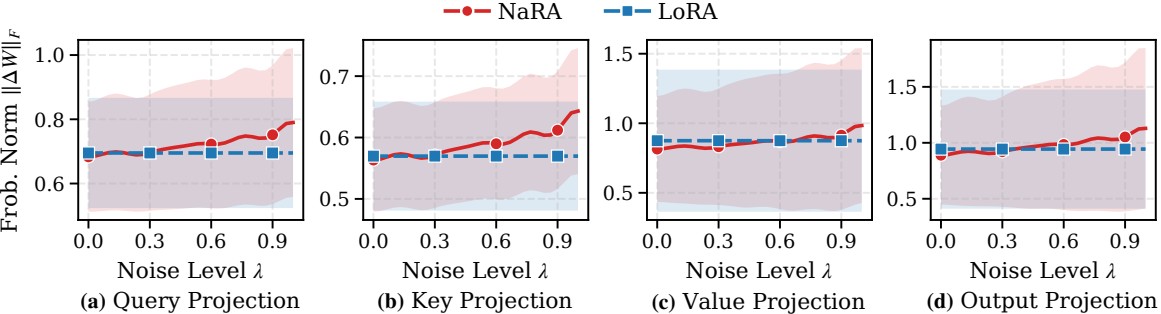

*Figure 3.* Variation of $\|\Delta W(\lambda)\|_F$ on code generation task. Solid lines and shading show the mean and standard deviation across layers.

higher average accuracy than the standard LoRA baseline reported in Table 2. Among the evaluated candidates, a scaling factor of 0.1 yields the superior average performance. The visualizations of the Frobenius norm presented in Appendix I also help clarify the underlying mechanisms. The factor $\eta$ directly influences the magnitude of the update matrices $\Delta W(\lambda)$ after training. Setting $\eta$ to 1.0 results in excessive fluctuations in $\Delta W(\lambda)$, leading to high variance. This volatility can destabilize the optimization trajectory. On the other hand, lowering $\eta$ to 0.01 dampens the noise-varying signal, keeping $\Delta W(\lambda)$ nearly constant and making the method behave like standard LoRA. We therefore use $\eta = 0.1$ by default to balance flexibility and stability.

### 6.2. Effectiveness of Gaussian Fourier Embedding

We evaluate three input embedding strategies for the hypernetwork: default Fourier embedding, a learnable MLP embedding, and a raw input baseline denoted as Scalar. As shown in Table 5, the Fourier embedding achieves superior overall performance, securing the highest average accuracy of 80.89% compared to the MLP (80.39%) and Scalar (80.26%) baselines. While the Scalar remains competitive on simpler arithmetic tasks, its performance degrades on complex reasoning benchmarks like AQuA, where the Fourier embedding maintains a distinct lead. Beyond average accuracy, our analysis reveals significant differences in training stability. The Fourier embedding demonstrates remarkable robustness, maintaining the lowest standard deviation across all tasks with an average of just ±0.34. In contrast, the baselines exhibit high volatility. Specifically, the MLP suffers from extreme instability on AQuA (±2.45), while the Scalar fluctuates notably on GSM8K (±1.17). These findings suggest that neither Scalar nor MLP can robustly distinguish noise levels, whereas high-frequency Fourier embedding effectively regularizes the hypernetwork to consistently capture the nuances across noise levels.

### 6.3. Impact of Hypernetwork Sharing Strategy

We investigate the optimal granularity for the hypernetwork configuration by comparing a global sharing strategy against

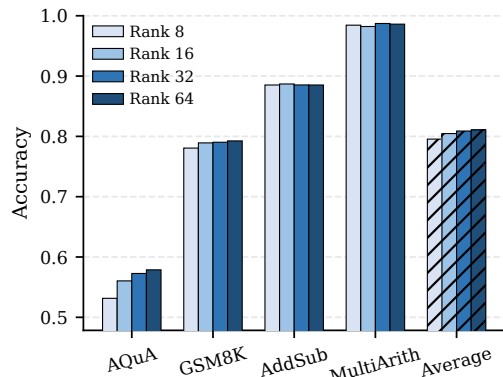

*Figure 4.* Performance of NaRA on mathematical reasoning tasks across different ranks, showing an increasing trend in average performance.

module-specific and grouped alternatives. The results presented in Table 7 demonstrate that the global sharing strategy consistently yields the highest average accuracy of 80.89% across the benchmarks. In contrast, the module-specific configurations all exhibit a performance drop, with accuracies ranging from 79.83% to 80.13%, despite possessing a greater number of trainable parameters. This empirical evidence suggests that the dependency on the noise level constitutes a global property of the diffusion process rather than a module-specific characteristic. Consequently, a single shared hypernetwork effectively captures this dependency and acts as a regularizer to prevent overfitting associated with increased architectural complexity.

### 6.4. Impact of Rank $r$

To investigate the sensitivity of NaRA to the low-rank dimension, we evaluate its performance across varying ranks $r \in \{8, 16, 32, 64\}$. To ensure training stability across different ranks, we scale the hidden and embedding dimensions of the hypernetwork accordingly, keeping the underlying structure identical across all configurations. Detailed hypernetwork configurations and their corresponding parameter counts for each rank are provided in Appendix J.

As illustrated in Figure 4, the average performance of NaRA scales positively with the rank $r$, a trend driven largely

*Table 5.* Ablation study on the noise-level embedding strategy (Fourier vs. MLP vs. Scalar) on mathematical reasoning tasks.

| METHOD | GSM8K | ADDSUB | AQUA | MULTIARITH | AVG |
|---|---|---|---|---|---|
| SCALAR | $78.77 \pm 1.17$ | $\textbf{88.69} \pm 0.89$ | $54.72 \pm 0.39$ | $\textbf{98.83} \pm 0.29$ | $80.26 \pm 0.68$ |
| MLP | $\textbf{79.59} \pm 0.80$ | $87.69 \pm 1.16$ | $\underline{55.57} \pm 2.45$ | $\underline{98.72} \pm 0.63$ | $\underline{80.39} \pm 1.26$ |
| FOURIER (OURS) | $\underline{79.03} \pm 0.24$ | $\underline{88.53} \pm 0.52$ | $\textbf{57.27} \pm 0.20$ | $\underline{98.72} \pm 0.42$ | $\textbf{80.89} \pm 0.34$ |

*Table 6.* Comparison with Multi-LoRA on mathematical reasoning. Mean and standard deviation are reported over three seeds.

| METHOD | PARAM(%) | GSM8K | ADDSUB | AQUA | MULTIARITH | AVG |
|---|---|---|---|---|---|---|
| MULTI-LORA | 1.68 | $77.18 \pm 1.07$ | $87.51 \pm 0.52$ | $37.79 \pm 1.38$ | $96.83 \pm 0.70$ | 74.83 |
| LORA | 0.42 | $\underline{78.96} \pm 0.42$ | $\underline{88.38} \pm 0.87$ | $\underline{54.45} \pm 1.90$ | $\underline{98.50} \pm 0.29$ | $\underline{80.07}$ |
| NARA (OURS) | 0.43 | $\textbf{79.03} \pm 0.24$ | $\textbf{88.53} \pm 0.52$ | $\textbf{57.27} \pm 0.20$ | $\textbf{98.72} \pm 0.63$ | $\textbf{80.89}$ |

*Table 7.* Ablation study on hypernetwork sharing strategies. *Shared* uses a single global hypernetwork. Rows with '/' denote separate hypernetworks (e.g., Q/K/V/O means four independent networks).

| STRATEGY | PARAM(%) | GSM8K | ADDSUB | AQUA | MULTIARITH | AVG |
|---|---|---|---|---|---|---|
| SHARED | 0.425 | $\underline{79.03}$ | $\textbf{88.53}$ | $\textbf{57.27}$ | $\underline{98.72}$ | $\textbf{80.89}$ |
| Q/K/V/O | 0.450 | 78.77 | 88.10 | 54.59 | $\textbf{99.06}$ | 80.13 |
| QV/KO | 0.434 | $\textbf{79.38}$ | $\underline{88.35}$ | 54.99 | 98.67 | $\underline{80.35}$ |
| QO/KV | 0.434 | 78.44 | 87.59 | $\underline{55.25}$ | 98.44 | 79.93 |
| QK/VO | 0.434 | 78.59 | 88.27 | 54.07 | 98.39 | 79.83 |

by significant improvements on the AQuA and GSM8K datasets. Notably, NaRA demonstrates remarkable parameter efficiency. Even at a reduced rank of $r = 16$, our method achieves an average performance of $0.8047$, surpassing the standard LoRA baseline at $r = 32$ which scores $0.8007$. This indicates that the noise-aware dynamic modulation of the update matrices allows for more effective fine-tuning in dLLMs than simply increasing the rank of static update matrices.

### 6.5. Comparison with Multi-LoRA

A natural alternative to NaRA is to train separate LoRA adapters for discrete noise-level intervals. We implement this Multi-LoRA baseline that partitions the noise range into 4 intervals with one dedicated adapter per interval. As shown in Table 6, Multi-LoRA performs worse than both LoRA and NaRA despite using $\sim 4\times$ more parameters. This supports our design motivation that discrete adapters prevent cross-noise-level information sharing, whereas NaRA's continuous shared subspace enables effective noise-aware adaptation with parameter efficiency.

### 6.6. Generalization to Image Diffusion

*Table 8.* NaRA on DreamBooth subject-driven generation with SDXL.

| METHOD | DINO ↑ | CLIP-I ↑ | FID ↓ |
|---|---|---|---|
| LORA | 0.0458 | 0.6901 | 22.13 |
| NARA (OURS) | **0.0610** | **0.7365** | **16.71** |

To demonstrate the broader applicability of NaRA, we eval-

uate it on the image diffusion domain using SDXL (Podell et al., 2024) under the DreamBooth subject-driven generation setup (Ruiz et al., 2023). Following the official HuggingFace `Diffusers` example, we use the `google/dreambooth` dataset, specifically the `dog6` subset comprising 5 images, and train for 1000 steps. During training, NaRA takes the normalized diffusion timestep $\lambda = t/T \in [0, 1]$ as the noise-level input to the hypernetwork. All other settings are kept identical to standard LoRA.

For evaluation, we generate images using DDIM with the prompt "a photo of sks dog", evaluated by DINO (Oquab et al., 2024), CLIP-I, and FID.

From Table 8, we can see that NaRA outperforms LoRA on all three metrics, demonstrating that noise-aware PEFT generalizes to the image diffusion domain.

## 7. Conclusion

In this work, we presented Noise-aware Low-Rank Adaptation (NaRA) to address the structural misalignment between static PEFT methods and the dynamic generation process of dLLMs. We identify that the requisite update matrices vary significantly across different noise levels, rendering noise-agnostic PEFT methods suboptimal for the multi-step denoising trajectory. NaRA resolves this limitation by incorporating a lightweight, globally shared hypernetwork that dynamically modulates the low-rank core matrix. This design enables the model to continuously adapt its behavior in accordance with the varying difficulty of the generation steps. Empirical results confirm that NaRA effectively aligns the adaptation mechanism with the underlying generative dynamics, achieving superior performance compared to static PEFT methods while maintaining negligible parameter overhead. Our findings establish NaRA as an effective PEFT framework tailored to dLLMs.

## Acknowledgements

This work was supported by National Natural Science Foundation of China under Grant no. 62136005 and Shenzhen fundamental research program JCYJ20250604144724032.

## Impact Statement

This paper presents work whose goal is to advance the field of Machine Learning. There are many potential societal consequences of our work, none which we feel must be specifically highlighted here.

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

# A. Proof of Theorem 4.1

In this section, we provide the constructive proof for Theorem 4.1. The proof proceeds by constructing a shared coordinate system derived from the union of the subspaces spanned by the target updates.

*Proof.* Let $\mathcal{U}_{col} = \bigcup_{i=1}^{N} \mathcal{C}(\Delta \mathbf{W}_i)$ denote the union of the column spaces of all target update matrices. We construct $\mathbf{B} \in \mathbb{R}^{d \times r}$ such that its columns form an orthonormal basis for $\mathcal{U}_{col}$. Since the columns of any specific update $\Delta \mathbf{W}_i$ reside entirely within $\mathcal{U}_{col}$, the projection of $\Delta \mathbf{W}_i$ onto the subspace spanned by $\mathbf{B}$ is lossless, implying $\mathbf{B}\mathbf{B}^\top \Delta \mathbf{W}_i = \Delta \mathbf{W}_i$.

Analogously, let $\mathcal{U}_{row} = \bigcup_{i=1}^{N} \mathcal{R}(\Delta \mathbf{W}_i)$ be the union of the row spaces. We construct $\mathbf{A} \in \mathbb{R}^{r \times k}$ such that its rows form an orthonormal basis for $\mathcal{U}_{row}$. As the rows of $\Delta \mathbf{W}_i$ lie within $\mathcal{U}_{row}$, the projection onto $\mathbf{A}$ is similarly lossless, satisfying $\Delta \mathbf{W}_i \mathbf{A}^\top \mathbf{A} = \Delta \mathbf{W}_i$.

With these global bases fixed, we define the dynamic core matrix for the $i$-th task as the projection of the update into this shared coordinate system: $\mathbf{C}_i = \mathbf{B}^\top \Delta \mathbf{W}_i \mathbf{A}^\top$. Substituting $\mathbf{C}_i$ back into the NaRA formulation recovers the original update matrix exactly:

$$\begin{aligned}
\mathbf{B}\mathbf{C}_i\mathbf{A} &= \mathbf{B}(\mathbf{B}^\top \Delta \mathbf{W}_i \mathbf{A}^\top)\mathbf{A} \\
&= (\mathbf{B}\mathbf{B}^\top)\Delta \mathbf{W}_i(\mathbf{A}^\top \mathbf{A}) \\
&= \Delta \mathbf{W}_i.
\end{aligned} \tag{12}$$

Here, the final equality holds because $\mathbf{B}\mathbf{B}^\top$ and $\mathbf{A}^\top \mathbf{A}$ act as identity operators on the column and row spaces of $\Delta \mathbf{W}_i$, respectively. Thus, the factorization is exact. □

# B. Training Algorithm Details

We present the training procedure of NaRA in Algorithm 1, where $\mathcal{U}(a, b)$ denotes the continuous uniform distribution on $[a, b]$, and for any sequence vector $\mathbf{v}$, $|\mathbf{v}|$ denotes its length while $\|\mathbf{m}\|_1$ denotes the number of masked tokens in the binary mask vector $\mathbf{m}$. The overall pipeline aligns with the standard SFT framework for diffusion dLLMs (Nie et al., 2025), incorporating our dynamic modulation mechanism into the forward pass.

---

**Algorithm 1** NaRA Supervised Fine-Tuning (SFT)

---

**Require:** Pre-trained model weights $\Theta$, Dataset $\mathcal{D}$, Hyperparameters.
**Require:** LoRA parameters $\Phi_{\text{LoRA}} = \{(\mathbf{A}_l, \mathbf{B}_l)\}_l$, Hypernetwork $\mathcal{F}_\phi$ parameterized by $\phi$.
 1: **Initialize:** Initialize $\Phi_{\text{LoRA}}, \mathcal{F}_\phi$ as described in Section 4.3.
 2: **repeat**
 3:     Sample batch $(\mathbf{p}, \mathbf{r}_0)$ from $\mathcal{D}$
 4:     Sample $t \sim \mathcal{U}(\varepsilon, 1)$ with $\varepsilon = 10^{-6}$
 5:     Apply masking to $\mathbf{r}_0$ via Eq. (1) to obtain binary mask vector $\mathbf{m}$ and masked response $\mathbf{r}_t$
 6:     Construct input $\tilde{x} \leftarrow [\mathbf{p}, \mathbf{r}_t]$
 7:     $\lambda \leftarrow \|\mathbf{m}\|_1 / |\mathbf{r}_0|$
 8:     Compute $\mathbf{C}(\lambda)$ via Eq. (7)
 9:     Broadcast $\mathbf{C}(\lambda)$ to all layers
10:     Compute adapter updates $\Delta \mathbf{W}$ via Eq. (4)
11:     Compute loss $\mathcal{L}$ via Eq. (2)
12:     Update $\Phi_{\text{LoRA}}, \phi$ via gradient descent
13: **until** convergence

---

# C. Impact of Block-wise Early Termination

In this section, we evaluate the acceleration capabilities of our proposed block-wise early termination strategy. We utilize the LLaDA-8B-Instruct model and conduct experiments across eight standard benchmarks. We report the total inference cost measured in GPU hours required to process the complete validation sets.

Crucially, our experiments confirm that this strategy preserves generation quality with negligible variance. Specifically, LoRA maintained identical accuracy across all benchmarks with early stopping. For NaRA, performance also remained

stable on most datasets, showing only a slight decrease on Winogrande from $0.8145$ to $0.8106$. Given the substantial efficiency gains, we consider this trade-off acceptable. Table 9 quantifies the reduction in computational cost. All time-measurement experiments were conducted on a single NVIDIA RTX 3090 GPU. Notably, the early termination mechanism reduces the total inference time of NaRA by approximately $81\%$, dropping from 39.17 to 7.47 GPU hours. The results demonstrate that our strategy significantly reduces inference overhead, enhancing the deployment efficiency of dLLMs.

*Table 9.* Comparison of total inference time measured in GPU hours on LLaDA-8B-Instruct validation sets. **Bold** text indicates minimum inference time.

| | LoRA (HOURS) | | NaRA (HOURS) | |
|---|---|---|---|---|
| | STANDARD | EARLY STOP | STANDARD | EARLY STOP |
| ARC-C | 1.4407 | 0.3206 | 1.5322 | **0.3054** |
| ARC-E | 2.8560 | 0.6335 | 3.1290 | **0.6078** |
| BOOLQ | 2.6483 | **0.5985** | 3.3589 | 0.6293 |
| HELLASWAG | 21.2611 | 4.7571 | 23.6502 | **4.5115** |
| OBQA | 0.5772 | 0.1278 | 0.6530 | **0.1224** |
| PIQA | 2.1707 | 0.5011 | 2.5411 | **0.4723** |
| SIQA | 2.2459 | 0.5113 | 2.5948 | **0.4933** |
| WINOGRANDE | 1.3978 | 0.3330 | 1.7098 | **0.3311** |
| **TOTAL** | 33.1571 | 7.7828 | 39.1691 | **7.4731** |

## D. Computational Efficiency Benchmarks

We conduct isolated timing benchmarks on identical inputs with lengths of 256 and 1024. We use 5 warmup passes and then run 50 repeated forward passes on an NVIDIA GeForce RTX 3090. For inference, we compare NaRA with both merged and unmerged LoRA. For training, weight merging is not applicable to LoRA, so we report only the unmerged setting.

*Table 10.* Per-step timing and peak VRAM for training and inference.

| | | TRAINING | | INFERENCE | |
|---|---|---|---|---|---|
| LENGTH | METHOD | STEP TIME (MS) | VRAM (GB) | STEP TIME (MS) | VRAM (GB) |
| | LoRA (MERGED) | — | — | 88.10 | 16.11 |
| 256 | LoRA (UNMERGED) | 248.89 | 18.21 | 92.88 | 16.18 |
| | NaRA (OURS) | 254.55 | 18.22 | 92.57 | 16.18 |
| | LoRA (MERGED) | — | — | 331.34 | 16.31 |
| 1024 | LoRA (UNMERGED) | 768.48 | 23.28 | 344.74 | 16.38 |
| | NaRA (OURS) | 770.23 | 23.30 | 344.96 | 16.38 |

Table 10 shows that NaRA adds only a small per-step overhead over unmerged LoRA. The extra cost is about 6 ms for training. Inference remains comparable. The VRAM increase is also negligible and stays below 0.02 GB. One limitation is that, unlike standard LoRA, NaRA cannot merge its weight update into the base weights. This leads to a small inference cost compared with merged LoRA, but the overhead remains acceptable.

## E. Additional Visualizations of Update Dynamics

To verify that NaRA's advantage is not merely due to a slight parameter budget difference, we evaluate LoRA at rank $r = 40$, which has a slightly larger parameter budget ($0.52\%$) than NaRA at $r = 32$ ($0.43\%$). All other settings are kept identical. We report mean and standard deviation over three random seeds.

## F. Comparison with Higher-Rank LoRA

To verify that NaRA's advantage is not merely due to a slight parameter budget difference, we evaluate LoRA at rank $r = 40$, which has a slightly larger parameter budget ($0.52\%$) than NaRA at $r = 32$ ($0.43\%$). All other settings are kept identical. We report mean and standard deviation over three random seeds.

As shown in Table 11, LoRA with rank 40 improves over rank 32 but still underperforms NaRA at rank 32, confirming that

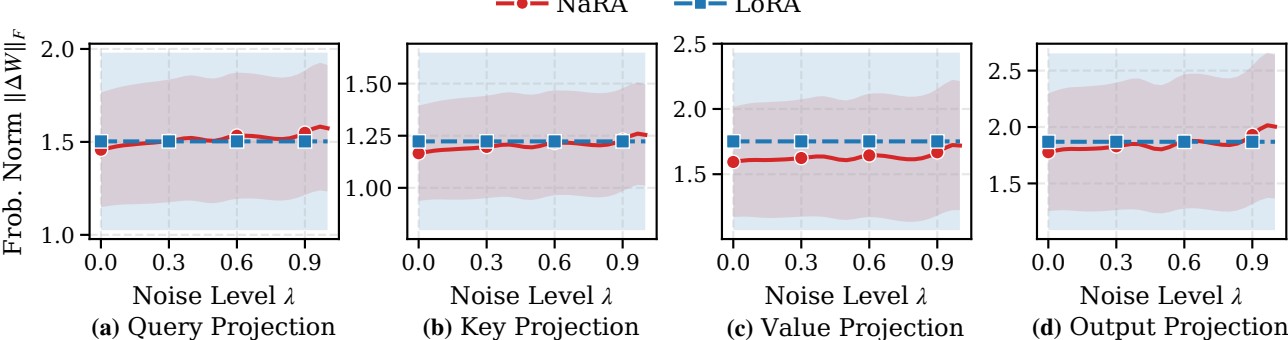

*Figure 5.* Variation of update Frobenius norms $\|\Delta W(\lambda)\|_F$ on the Math task. Solid lines and shaded regions denote the mean and standard deviation across layers, respectively.

*Table 11.* Comparison of LoRA (r=32), LoRA (r=40), and NaRA (r=32) on mathematical reasoning benchmarks. Mean and standard deviation are reported over three seeds.

| METHOD | RANK | PARAM(%) | GSM8K | ADDSUB | AQUA | MULTIARITH | **AVG** |
|---|---|---|---|---|---|---|---|
| LoRA | 40 | 0.52 | **79.08 ± 0.74** | 87.17 ± 0.73 | 55.91 ± 2.23 | 98.61 ± 0.29 | 80.19 |
| LoRA | 32 | 0.42 | 78.96 ± 0.42 | 88.38 ± 0.87 | 54.45 ± 1.90 | 98.50 ± 0.29 | 80.07 |
| NARA (OURS) | 32 | 0.43 | 79.03 ± 0.24 | **88.53 ± 0.52** | **57.27 ± 0.20** | **98.72 ± 0.63** | **80.89** |

NaRA's gains stem from noise-aware dynamic adaptation rather than a parameter budget advantage.

## G. Detailed Training Settings

In this section, we provide a comprehensive account of the implementation details to facilitate reproducibility. We implement standard baselines using the Hugging Face PEFT library, while HiRA is evaluated using its official repository to ensure strict alignment with the original method.

For our proposed NaRA, the specific hyperparameter configurations are summarized in Table 12. These settings are consistently applied across all models and datasets, with one notable exception: for the Commonsense170k dataset, the learning rate is adjusted to 1e-5.

Structurally, NaRA applies adapters to the query, key, value, and output projection layers. The dynamic modulation matrix is generated by a hypernetwork taking Fourier-embedded noise levels as input. It is implemented as an MLP with two hidden layers and one output layer, using SiLU activations. Please refer to Table 12 for the specific hidden dimensions.

Regarding optimization, all models are trained using AdamW with FP16 mixed-precision to enhance computational efficiency. We conduct all fine-tuning experiments on NVIDIA GeForce RTX 3090 GPUs, each equipped with 24GB of VRAM. To strictly manage memory consumption within this capacity, we set the per-device batch size to 1 and employ 32 gradient accumulation steps, resulting in an effective global batch size of 32. Training consists of 10 epochs for mathematical reasoning and 1 epoch for other tasks.

## H. Inference Configuration

In this section, we detail the inference hyperparameters used for evaluating NaRA. We utilize a Semi-Autoregressive (Semi-AR) decoding strategy for all tasks.

To balance generation quality and efficiency, we adjust the Answer Length, Block Size, and inference steps. Consistent with our experimental design, the number of inference steps is strictly set equal to the defined Answer Length for all evaluations.

Table 13 summarizes the specific configurations for each dataset. To ensure a fair comparison, we largely follow the inference settings from (Nie et al., 2025). Notably, while most benchmarks are evaluated in a zero-shot setting to test the model's direct instruction-following ability, we apply 3-shot prompting for MBPP to mitigate the performance degradation observed in zero-shot scenarios for this specific task.

*Table 12.* Hyperparameter configurations for NaRA fine-tuning.

| Hyperparameter | Value |
|---|---|
| *Adapter Architecture* | |
| Rank ($r$) | 32 |
| Target Modules | $q\_proj, k\_proj, v\_proj, o\_proj$ |
| Dropout | 0.05 |
| Scaling Factor ($\eta$) | 0.1 |
| *Hypernetwork Configuration* | |
| Input Mode | Noise Level |
| Embedding Type | Fourier |
| Embedding Dimension | 64 |
| MLP Hidden Sizes | [256, 512] |
| Initialization Strategy | Zero-last |
| *Optimization* | |
| Optimizer | AdamW |
| Learning Rate | 1e-4 (Commonsense170k: 1e-5) |
| Warmup Ratio | 0.05 |
| Per-Device Batch Size | 1 |
| Gradient Accumulation | 32 |
| Effective Batch Size | 32 |
| Mixed Precision | FP16 |

# I. Visualization of Dynamic Matrix Norms under Varying Scaling Factors $\eta$

In this section, we provide visual evidence supporting the ablation study on the scaling factor $\eta$. We plot the Frobenius norm of the dynamic matrix $C$ across different noise levels to analyze the training dynamics.

Figure 6 illustrates the behavior when $\eta = 0.01$. The norm remains nearly flat and indicates that the dynamic modulation is negligible. This confirms that the model behaves like a static LoRA and fails to utilize time-dependent information.

Figure 7 displays the behavior when $\eta = 1.0$. The norm exhibits high variance and sharp spikes. These fluctuations indicate unstable parameter updates that hinder the convergence of the diffusion model.

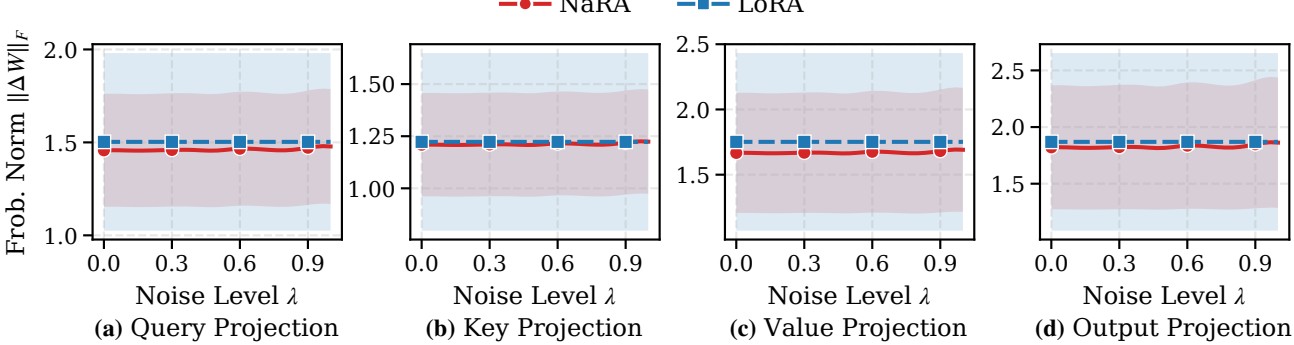

*Figure 6.* Visualization of the update Frobenius norms $\|\Delta W(\lambda)\|_F$ with $\eta = 0.01$. The trajectory is nearly constant and indicates a degeneration to static behavior.

# J. Hypernetwork Configurations for different ranks

To maintain consistency and stability during training across different rank settings, we modulate the width of the hypernetwork. The specific dimensions and the relative parameter overhead are summarized in Table 14.

*Table 13.* Inference hyperparameters across different domains. Steps are set equal to the Answer Length.

| Domain / Task | Answer Length | Block Size | Shots |
|---|---|---|---|
| *Commonsense Reasoning* | | | |
| All 8 Sub-tasks | 64 | 4 | 0 |
| *Mathematical Reasoning* | | | |
| GSM8K | 256 | 8 | 0 |
| AddSub | 256 | 8 | 0 |
| AQuA | 256 | 8 | 0 |
| MultiArith | 128 | 8 | 0 |
| *Code Generation* | | | |
| HumanEval | 512 | 32 | 0 |
| MBPP | 512 | 32 | 3 |

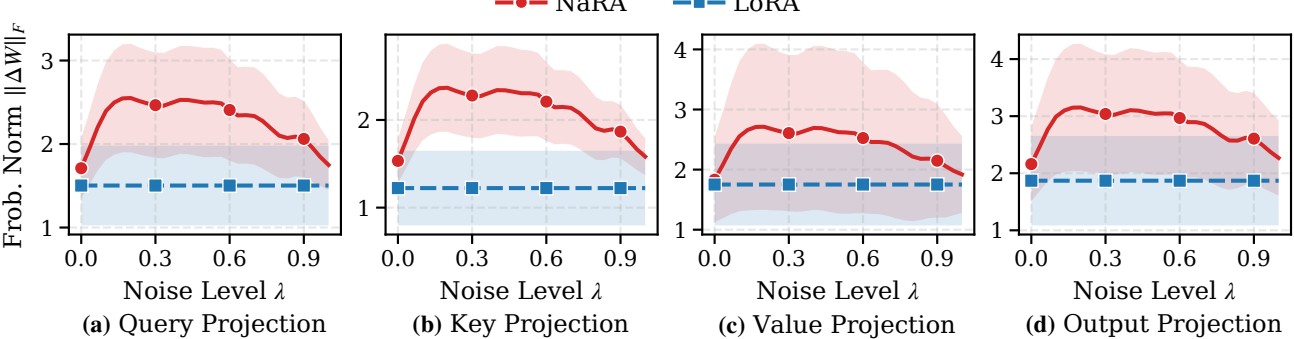

*Figure 7.* Visualization of theupdate Frobenius norms $\|\Delta W(\lambda)\|_F$ with $\eta = 1.0$. The trajectory shows drastic fluctuations and indicates optimization instability.

## K. Dataset Details

In this section, we provide a comprehensive description of the datasets we use.

### K.1. Commonsense Reasoning

For the commonsense reasoning domain, we utilize the Commonsense170k dataset for fine-tuning. This dataset aggregates eight diverse sub-tasks, combining a total of 170,420 query-answer pairs. The evaluation suite covers a wide range of capabilities. BoolQ is a question-answering dataset for yes/no questions containing naturally occurring queries. PIQA focuses on physical commonsense reasoning about everyday situations, while SIQA benchmarks reasoning about social interactions and emotional implications. HellaSwag tests commonsense natural language inference by requiring the model to complete sentences describing everyday events. Winogrande measures robust capabilities in resolving ambiguous pronouns. For science-related reasoning, we use ARC-Challenge and ARC-Easy, where the former specifically targets questions difficult for retrieval-based methods. Finally, OpenBookQA requires multi-step reasoning using open-book general knowledge.

*Table 14.* Hypernetwork architectural configurations and parameter efficiency across different ranks $r$. We report the dimensions of the embedding layer ($d_{emb}$) and hidden layers ($d_{h1}, d_{h2}$), along with the percentage of trainable parameters.

| RANK ($r$) | $d_{emb}$ | $d_{h1}$ | $d_{h2}$ | PARAM(%) |
|---|---|---|---|---|
| 8 | 4 | 16 | 32 | 0.1046 |
| 16 | 16 | 64 | 128 | 0.2094 |
| 32 | 64 | 256 | 512 | 0.4252 |
| 64 | 128 | 512 | 1024 | 0.8890 |

## K.2. Mathematical Reasoning

We employ the Math14k dataset for training mathematical reasoning capabilities. Distinguished from previous iterations, this dataset specifically excludes samples from MAWPS to strictly prevent data contamination with evaluation benchmarks such as AddSub and MultiArith. Instead, it aggregates training samples from GSM8K and AQuA, enriched with high-quality rationales synthesized by ChatGPT and GPT-4. We evaluate performance on four benchmarks. GSM8K contains high-quality linguistically diverse grade school math word problems. AddSub focuses on arithmetic word problems involving addition and subtraction. AQuA consists of algebraic word problems with rationales, requiring complex logical reasoning. MultiArith includes math word problems that require multiple steps of arithmetic operations.

## K.3. Code Generation

For code generation, we utilize the pre-processed version of the CodeFeedback dataset released by (Meng et al., 2024), which has been filtered to retain 104,848 Python-related samples. Building upon this subset, we further conserve computational resources by selecting only those samples with a sequence length of 512 tokens or fewer. This results in a final high-quality dataset of 48,799 examples used for our fine-tuning. Evaluation is performed on HumanEval and MBPP benchmarks to assess the model's proficiency in generating syntactically correct and functional Python code.

## L. Experimental Settings for Loss-Noise Analysis

We conduct the loss landscape analysis presented in Figure 1 using LLaDA-Instruct (Nie et al., 2025) fine-tuned on the Math14k dataset (Hu et al., 2023), as described in Section 5.3. The evaluation is performed on the test split of the AQuA dataset. For each data instance, we simulate the diffusion corruption process by uniformly sampling a noise level $\lambda \in [0, 1]$. We determine the number of tokens to mask as $m = \lfloor \lambda \cdot L_s \rfloor$, where $L_s$ denotes the length of the response segment, and mask $m$ randomly selected tokens. We then compute the cross-entropy loss over these masked tokens. We repeat this corruption and evaluation process four times for each sample. The resulting noise level and loss pairs are visualized via a scatter plot, overlaid with a Locally Weighted Scatterplot Smoothing (LOWESS) trend line using a smoothing fraction of 0.5.

## M. Combination of NaRA with DoRA

NaRA is designed as a plug-and-play module that can be composed with other PEFT methods. We instantiate NaRA on top of DoRA (Liu et al., 2024) by introducing a $\lambda$-conditioned scaling matrix into DoRA's magnitude-direction decomposition. Results on code generation tasks are shown in Table 15.

*Table 15.* Combining NaRA with DoRA on code generation. All methods share the same rank r = 32. The "Param" column indicates the percentage of trainable parameters relative to the base model. **Bold** indicates the best result, and underline indicates the second best.

| METHOD | PARAM(%) | HUMANEVAL | MBPP | AVG |
|---|---|---|---|---|
| DORA | 0.423 | 39.02 | 36.70 | 37.86 |
| DORA+NARA (OURS) | 0.432 | **39.84** | **38.47** | **39.16** |

The results confirm that NaRA's noise-aware modulation remains effective when combined with DoRA, indicating that NaRA can serve as a general noise-awareness module for various PEFT methods.

*Table 16.* NaRA-C ablation on mathematical reasoning. All methods share the same rank r = 32. **Bold** indicates the best result, and underline indicates the second best.

| METHOD | GSM8K | ADDSUB | AQUA | MULTIARITH | AVG |
|---|---|---|---|---|---|
| LORA | 78.96 | 88.38 | 54.45 | 98.50 | 80.07 |
| NARA-C | 78.43 | **89.12** | 55.12 | 98.50 | 80.29 |
| NARA (OURS) | **79.03** | 88.53 | **57.27** | **98.72** | **80.89** |

# N. Ablation on NaRA-C

We ablate the necessity of noise conditioning by introducing NaRA-C, a variant in which the core matrix $\mathbf{C}$ is learnable but not conditioned on $\lambda$. All other settings are identical to NaRA.

*Table 17.* NaRA-C ablation on commonsense reasoning. All methods share the same rank r = 32. **Bold** indicates the best result, and underline indicates the second best.

| METHOD | ARC-C | ARC-E | BOOLQ | HELLA | OBQA | PIQA | SIQA | WINO | AVG |
|---|---|---|---|---|---|---|---|---|---|
| LoRA | 75.74 | 87.84 | **68.02** | 90.52 | 77.87 | **85.40** | 75.38 | 80.69 | 80.18 |
| NARA-C | 79.92 | 88.62 | 55.61 | 75.98 | 74.00 | 83.64 | 63.13 | 24.13 | 68.13 |
| NARA (OURS) | **85.92** | **95.65** | 67.31 | **90.89** | **86.87** | 85.18 | **79.82** | **81.06** | **84.09** |

As shown in Table 16 and Table 17, NaRA-C is slightly better than LoRA on math tasks but still worse than NaRA, and substantially underperforms both LoRA and NaRA on commonsense tasks. This confirms that noise-level conditioning—not merely an additional learnable matrix—is the key driver of NaRA's gains.

# O. Qualitative Examples

We present representative examples of NaRA's outputs, including both image and text results.

**Image Results.** Figure 8 shows example images generated under the DreamBooth subject-driven generation setup.

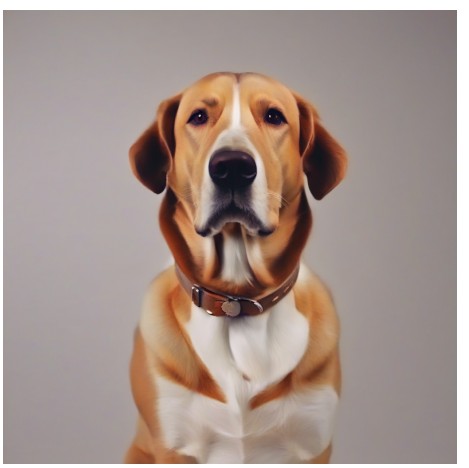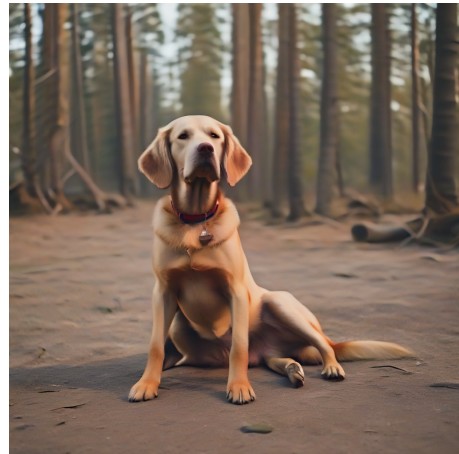

*Figure 8.* Example DreamBooth images of the `dog6` subject generated by NaRA on SDXL.

**Code Results.** *Task*: count how many times a substring appears in a string, *including overlapping* occurrences.

**LLaDA+NaRA**

```
count = 0
for i in range(n - m + 1):              # advances by 1: all positions checked
    if string[i:i + m] == substring:
        count += 1
return count
# how_many_times('aaaa', 'aa') -> 3   (correct: positions 0, 1, 2)
```

**LLaDA+LoRA**

```
count = 0
i = 0
while i < len(string):
    if string[i:i+len(substring)] == substring:
        count += 1
```

```
        i += len(substring)              # skips len(sub) positions: overlaps missed
    return count
    # how_many_times('aaaa', 'aa') -> 2  (incorrect: only positions 0, 2)
```

**Math Results.** *Task*: Emil is 19 years old. When he turns 24, he will be half his dad's age and twice his brother's age. What is the *current* sum of his dad's and brother's ages? (Answer: 50)

> **LLaDA+NaRA**
>
> When Emil turns 24, he will be $24 - 19 = 5$ years older than now. At age 24: his dad will be $24 \times 2 = 48$ and his brother $24/2 = 12$. Their *current* ages are $48 - 5 = 43$ and $12 - 5 = 7$. Sum $= 43 + 7 = \mathbf{50}$. ✓

> **LLaDA+LoRA**
>
> When Emil turns 24, he will be $19 + 24 = 43$ years old. His dad will then be $43 \times 2 = 86$ and his brother $43/2 \approx 21$. Sum $= 86 + 21 = \mathbf{107}$. ✗

*Task*: Jim has a 20-pack of gum. He chews 1 piece per 2 hours over an 8-hour school day, 1 piece on the way home, and 1 piece after dinner. He then gives half of his remaining gum to his sister. How many pieces does Jim have left? (Answer: 7)

> **LLaDA+NaRA**
>
> At school: $8/2 = 4$ pieces chewed.
> On way home and after dinner: $1 + 1 = 2$ pieces chewed.
> Remaining before giving to sister: $20 - 4 - 2 = 14$ pieces.
> Given to sister: $14/2 = 7$ pieces.
> Jim has $20 - 4 - 2 - 7 = \mathbf{7}$ pieces left. ✓

> **LLaDA+LoRA**
>
> At school: $8/2 = 4$ pieces chewed.
> On way home and after dinner: $1 + 1 = 2$ pieces chewed.
> He gives away $20 - 4 - 2 = 14$ pieces to his sister.
> Jim has 14 pieces of gum left. ✗

