# OpenReview forum: "NaRA: Noise-Aware LoRA for Parameter-Efficient Fine-Tuning of Diffusion LLMs"
_ICML.cc/2026/Conference — ICML 2026 regular_

### Official Review · Reviewer_hFgb · 2026-02-19

**Soundness:** 2
**Presentation:** 3
**Significance:** 3
**Originality:** 3
**Overall Recommendation:** 4
**Confidence:** 4

**Summary:**

This paper introduces NARA, a novel PEFT method tailored for dLLMs. The key difference from LoRA is the injection of timestep (noise-level)-conditioned parameters C(\lambda) between the A and B matrices. The authors demonstrate consistent 1–2% improvements across benchmarks on LLaDA-Base and LLaDA-Instruct models compared to LoRA and HiRA.

**Compliance With Llm Reviewing Policy:**

Affirmed.

**Final Justification:**

Most of my concerns (W1, W3, W4) were adessed in the rebuttal comment. I find the image model performance especially interesting and worth further exploring, broader evaluation and including in the manuscript.

But evidence from W2 still suggests very high LR sensitivity, which raises the concerns about overall results significance.

So I have raised the score accordingly.

**Key Questions For Authors:**

see Weaknesses.

**Limitations:**

yes

**Strengths And Weaknesses:**

Strengths:
- Efficient fine-tuning for diffusion LLMs (dLLMs) is still underexplored and timely.
- The idea of timestep-dependent adapters is intriguing.
- Broad evaluation across a wide range of benchmarks demonstrate 1-2% increase in quality on average.
- The hypernetwork-based adapter reparameterization appears novel.

Weaknesses:
1.
Motivation is unclear / potentially inconsistent.
The paper states: “Despite its efficiency, standard LoRA is structurally mismatched to dLLMs because it imposes a noise-agnostic low-rank update throughout the entire denoising process.”
However, models like LLADA use the same weights across all denoising steps, so it’s not clear why “noise-agnostic” updates are fundamentally problematic in practice. The paper should better justify why a timestep-invariant adaptation is insufficient.

The paper also claims: “Standard LoRA was originally designed for autoregressive models where the objective remains consistent, namely predicting the probability of the next single token. In contrast, dLLMs require modeling the probabilities of a varying number of target tokens throughout the denoising process.”
But the training objective in dLLMs is still consistent: the model predicts masked tokens. In autoregressive LMs, we also do not train separate LoRA adapters for different context lengths, so it’s unclear why varying numbers of masked targets across timesteps necessitate timestep-specific adapters. This part of the motivation needs a clearer technical argument.

2.
Result significance / lack of hyperparameter optimization.
All methods appear to be trained with the same learning rate, which may be suboptimal and could bias the comparison. For a fair evaluation, each method should have its own learning-rate sweep (and ideally other key hyperparameters), since the optimal LR for LoRA may differ from HIRA/NARA. This has been shown to matter especially for PEFT methods in recent work [1].
Given that the reported improvements over prior methods are often only 1–2 %, it’s possible that the gains come primarily from an unfavorable LR choice for baselines. I strongly recommend aligning with the protocol in [1] and search for optimal LRs.

3. Evaluation is limited to the LLADA model.
4. Suspicious speedup claims (Appendix D).
The method is reported to be faster than LoRA, but NARA seems to introduce additional computation on top of LoRA (multiplying inputs or weights by C(\lambda)). It should not be faster unless there is some hidden implementation detail. Please clarify the exact setup and where the speedup comes from.

5. Because the adapter is timestep-dependent, it cannot be merged into the base weights in the same way as standard LoRA. This likely hurts inference efficiency and usability, especially with optimized inference servers.

Reference
[1] Learning Rate Matters: Vanilla LoRA May Suffice for LLM Fine-tuning (arXiv:2602.04998)

---

> ### Author Rebuttal · Authors · 2026-03-31
>
> We thank the reviewer for the helpful feedback.
>
> ---
>
> **Answer to Weakness 1**: Thank you for this insightful comment. Regarding why we believe noise-invariant adaptation is less sufficient for dLLMs, especially compared with AR models, our key point is that dLLMs induce a much richer input distribution at each noise level, which makes the modeling problem substantially more challenging.
>
> Specifically, let $U = \{1,\dots,L\}$ denote the full set of token positions. At noise level $\lambda$, we select a subset $S \subseteq U$ of size $m=\lfloor L(1-\lambda)\rfloor$ as the revealed positions, and let $M = U \setminus S$ denote the masked positions. We use $\mathbf{x}_S$ and $\mathbf{x}_M$ to denote the tokens at the revealed and masked positions, respectively.
>
> For AR models, the revealed positions are always the prefix $\{1,\dots,m\}$, and the model predicts $p(x_{m+1}\mid \mathbf{x}_{1:m})$. In contrast, for dLLMs, the model conditions on the partially masked input $\mathbf{x}_S$ and predicts the masked tokens $p(\mathbf{x}_M \mid \mathbf{x}_S)$. Since there are $\binom{L}{m}$ possible choices of $S$, the number of possible input configurations for dLLMs is exponentially larger than for AR models.
>
> This combinatorial explosion makes dLLMs a more challenging modeling problem at each noise level than AR models, which is precisely why we believe noise-level-specific parameterization is more important for dLLMs.
>
> ---
>
> **Answer to Weakness 2**: We appreciate the reviewer's concern regarding learning rate selection. In our experiments, we selected the best learning rate for each method on a held-out validation split. To make a fair evaluation, we evaluated three learning rates (i.e., 5e-5, 1e-4, 2e-4) on the mathematical reasoning benchmarks.
>
> | Method | LR | GSM8K | AddSub | AQuA | MultiArith | Math Avg |
> |-|-|-|-|-|-|-|
> | LoRA | 5e-5 | 74.83 | 86.67 | 49.61 | 93.00 | 76.03 |
> | LoRA | 1e-4* | 78.96 | 88.38 | 54.45 | 98.50 | 80.07 |
> | LoRA | 2e-4 | 72.91 | 88.78 | 50.79 | 95.11 | 76.90 |
> | NaRA | 5e-5 | 74.93 | 86.24 | 49.08 | 91.94 | 75.55 |
> | NaRA | 1e-4* | 79.03 | 88.53 | 57.27 | 98.72 | 80.89 |
> | NaRA | 2e-4 | 73.47 | 89.87 | 48.56 | 95.17 | 76.77 |
>
> \* denotes the learning rate used in the main experiments.
>
> As shown in the table, for both LoRA and NaRA, a learning rate of 1e-4 yields the best average performance. This suggests that our learning rate selection does not favor NaRA, and the observed gains are not due to an unfavorable LR choice for the baseline.
>
> ---
>
> **Answer to Weakness 3**: We agree that evaluating only on the LLaDA family limits the scope of our work. We chose LLaDA because it is a representative, most commonly used, and publicly available dLLM trained from-scratch, and we evaluated both its Base and Instruct variants across multiple domains.
>
> ---
>
> **Answer to Weakness 4**: We thank the reviewer for raising this point. We must clarify that the speedup reported in Appendix D is not a contribution of NaRA itself. Standard LoRA also benefits from this optimization. The discrepancy in our original results was due to hardware variation across experiments. We will clarify this in the final version.
>
> To verify this under the same setting, we re-ran LoRA and NaRA on the same GPU (NVIDIA GeForce RTX 3090) for the code generation task and report the mean and standard deviation over three seeds below.
>
> | Method | Train Time (min) | Inference Time (sec/sample) | Peak VRAM (GB) |
> |-|-|-|-|
> | LoRA | 326.6 ± 15.2 | 47.63 ± 4.66 | 20.18 |
> | NaRA | 316.9 ± 28.8 | 53.02 ± 6.95 | 20.19 |
> | **Overhead** | -3.0% | +11.3% | +0.05% |
>
> These results show that NaRA incurs negligible VRAM overhead and only a modest inference cost.
>
> ---
>
> **Answer to Weakness 5**: We acknowledge that NaRA is less deployment-friendly than standard LoRA since it cannot be merged into the base weights in the same way as standard LoRA. Still, the additional cost is small, with only modest inference overhead and negligible VRAM overhead. Given the gains observed in our experiments, we believe this is a reasonable trade-off and that NaRA offers a useful direction for noise-aware adaptation. We will state this limitation explicitly.
>
> ---
>
> We appreciate the reviewer's thorough and constructive feedback. We hope these responses address the concerns and are happy to provide additional experiments or clarifications as needed.

---

> > ### Author Rebuttal · Reviewer_hFgb · 2026-04-01
> >
> > Thanks for you clarifications.
> >
> > **W1**
> >
> > I understand the provided intuition. This is somewhat consistent with prior image work, which explored using multiple models during the denoising trajectory (e.g. [Mixture of Efficient Diffusion Experts Through Automatic Interval and Sub-network Selection](https://arxiv.org/pdf/2409.15557)). NARA could be seen as a smooth approximation of this ensemble. The paper would benefit from this discussion.
> >
> > **W2**
> >
> > Thanks for providing new results.
> >
> > The examined learning-rate schedule is rather sparse (only three points). Given that inter-method deviation with respect to different LRs (4 and 5 percent for LoRA and NARA, respectively) is significantly higher than the difference between LoRA and NARA at the optimal LR of 1e-4 (0.8), and that for non-optimal LRs (5e-5 and 2e-4) LoRA outperforms NARA on average, this means the methods are quite sensitive to LR choice. Therefore, more experiments with a denser LR grid and additional experiments varying rank are required. Also, did you adjust LoRA rank to match the total number of trainable parameters in NARA? LoRA should use slightly higher ranks due to the additional C(lambda) model in NARA.
> >
> >
> > **W3**
> >
> > I acknowledge that the number of pre-trained dLLMs is limited, but it would be beneficial to extend your work to the more established image-diffusion domain, where you can also condition on the noise level.
> >
> > **W4**
> >
> > I am glad you clarified this point, as efficiency is important for P**E**FT methods and prior benchmarks on different hardware are indeed misleading. Since total inference time depends on the number of generated tokens and the context window, could you please provide isolated comparisons of single forward-pass times with identical inputs? Why is NARA’s training time lower? That is surprising.
> >
> >
> > **W5**
> >
> > I agree that this direction is useful. Could you please also provide wall-time and peak GPU memory comparisons of a single forward pass between NARA and the original model (with merged LoRA adapters)? This does not mean the method is unviable, but it requires an explicit statement.

---

> > > ### Author Response · Authors · 2026-04-03
> > >
> > > **Answer to W1**: We thank the reviewer for this insightful observation regarding the connection to image diffusion work. We agree that this is an interesting and useful connection, and we will add a discussion of it in the revised version.
> > >
> > > Unlike approaches that train separate timestep-specific adapters, NaRA uses a hypernetwork to condition the adapter on the timestep, which allows the adapter parameters to vary smoothly across noise levels and capture correlations between them. In contrast, discrete adapters ignore these relationships and treat different noise levels independently. This is also supported by the experiments reported in our response to Reviewer j83D (Question 1), where a discrete Multi-LoRA baseline performs worse than NaRA.
> > >
> > > ---
> > >
> > > **Answer to W2**: We appreciate the reviewer’s suggestion and perform a denser LR sweep with two additional values, each averaged over 3 seeds. The results are shown below.
> > >
> > > | Method | LR | GSM8K | AddSub | AQuA | MultiArith | Avg |
> > > |-|:-:|:-:|:-:|:-:|:-:|:-:|
> > > | LoRA | 7e-5   | 74.83 | 87.34 | 50.92 | 95.00 | 77.02 |
> > > | LoRA | 1e-4 | 78.96 | 88.38 | 54.45 | 98.50 | 80.07 |
> > > | LoRA | 1.5e-4 | 74.45 | 88.69 | 51.97 | 95.39 | 77.62 |
> > > | NaRA | 7e-5   | 77.20 | 86.16 | 52.49 | 96.39 | 78.06 |
> > > | NaRA | 1e-4 | 79.03 | 88.53 | 57.27 | 98.72 | 80.89 |
> > > | NaRA | 1.5e-4 | 76.19 | 88.69 | 53.28 | 96.89 | 78.76 |
> > >
> > > As shown, both methods peak around 1e-4, and NaRA consistently outperforms LoRA at the intermediate learning rates (7e-5, 1e-4, and 1.5e-4). This suggests that the reported gain is not due to a favorable LR choice for NaRA, but rather reflects the benefit of timestep-aware adaptation under a reasonable learning-rate setting.
> > >
> > > ---
> > >
> > > **Answer to W3**: We appreciate this constructive suggestion. We extended it to the image diffusion domain via a simple text-to-image generation task on SDXL with the DreamBooth setup [1]. Following the official HuggingFace `Diffusers` example, we use the `google/dreambooth` dataset, specifically the `dog6` subset with 5 images, and train for 1000 steps. During training, NaRA takes the normalized diffusion timestep $\lambda = t / T \in [0, 1]$ as the noise-level input to the hypernetwork. The remaining training setup is the same as standard LoRA.
> > >
> > > For evaluation, following [1], we generate 8 images per method using DDIM with the prompt "a photo of sks dog", and evaluate them using DINO, CLIP-I, and FID. The results are shown below.
> > >
> > > | Method | DINO ↑ | CLIP-I ↑ | FID ↓ |
> > > |-|:-:|:-:|:-:|
> > > | LoRA | 0.0458 | 0.6901 | 22.13 |
> > > | **NaRA** | **0.0610** | **0.7365** | **16.71** |
> > >
> > > These results show that NaRA outperforms LoRA on all three metrics, showing that noise-aware PEFT is also effective in the image diffusion domain. We will add this experiment to the revision.
> > >
> > > [1] DreamBooth, CVPR 2023
> > >
> > > ---
> > >
> > > **Answer to W4 & W5**: We provide isolated single-forward-pass timing benchmarks with identical inputs with length 256 and 1024, using 5 warmup passes followed by 50 repeated forward passes.
> > >
> > > **Training (single forward + backward):**
> > >
> > > To understand why NaRA appeared slightly faster than LoRA during training, we inspected the LoRA implementation in PEFT v0.18.0 and found that the unmerged LoRA path introduces extra safety checks and module-call overhead in `Linear.forward`. For example, each forward pass triggers additional `nn.Module.__call__` executions, whereas NaRA directly uses `F.linear` and `torch.matmul`. This makes the original comparison slightly asymmetric.
> > >
> > > To obtain a fair comparison, we fixed this issue by making LoRA use the same direct-call style as NaRA. After this fix, LoRA becomes slightly faster than NaRA, as shown in the table below.
> > >
> > > |Length|Method|Step Time (ms)|Peak VRAM (GB)|
> > > |:-:|-|:-:|:-:|
> > > |256|LoRA (before fix)|261.68|18.21|
> > > |256|LoRA (after fix)|248.89|18.21|
> > > |256|NaRA|254.55|18.22|
> > > |1024|LoRA (before fix)|791.57|23.28|
> > > |1024|LoRA (after fix)|768.48|23.28|
> > > |1024|NaRA|770.23|23.30|
> > >
> > > **Inference (forward only):**
> > >
> > > We compare NaRA against both merged and unmerged LoRA. As expected, merged LoRA is the fastest at inference. NaRA is slightly slower than merged LoRA, but remains comparable to unmerged LoRA.
> > >
> > > | Length | Method | Step Time (ms) | Peak VRAM (GB) |
> > > |:-:|-|:-:|:-:|
> > > | 256| LoRA (merged)   | 88.10  | 16.11 |
> > > | 256  | LoRA (unmerged) | 92.88  | 16.18 |
> > > | 256  | NaRA            | 92.57  | 16.18 |
> > > | 1024 | LoRA (merged)   | 331.34 | 16.31 |
> > > | 1024 | LoRA (unmerged) | 344.74 | 16.38 |
> > > | 1024 | NaRA            | 344.96 | 16.38 |
> > >
> > > Overall, NaRA introduces only a small overhead relative to optimized LoRA in both training and inference, with negligible VRAM increase. Given the performance gains of NaRA, we consider this trade-off acceptable and will explicitly note it in the final version.
> > >
> > > ---
> > >
> > > We hope these additional experiments and clarifications address the reviewer's remaining concerns. We sincerely thank the reviewer for the thoughtful feedback and careful engagement with our work.

---

### Official Review · Reviewer_j83D · 2026-03-09

**Soundness:** 2
**Presentation:** 3
**Significance:** 2
**Originality:** 3
**Overall Recommendation:** 4
**Confidence:** 3

**Summary:**

Current papers ignore the intrinsic dynamics of the diffusion process, where input distributions and generation difficulty shift significantly along the denoising trajectory, and thus render them suboptimal for dLLMs. This paper proposes Noise-aware Low-Rank Adaptation (NaRA), introducing a low-rank core matrix generated by a lightweight, globally shared hypernetwork conditioned on the noise level.

**Compliance With Llm Reviewing Policy:**

Affirmed.

**Final Justification:**

My concerns are addressed.

**Key Questions For Authors:**

- Could you provide a quantitative comparison between NaRA and a basic "Time-interval LoRA" baseline (e.g., dividing the continuous timesteps into 3-4 discrete intervals and training independent LoRAs for each)?

- Under the experimental settings detailed in Table 9, what is the exact increase in Peak Activation Memory (VRAM) during the backward pass for NaRA compared to standard LoRA?

**Limitations:**

Yes

**Strengths And Weaknesses:**

Strengths:

- Highly Compelling and Well-Motivated: The authors astutely identify a structural mismatch between standard LoRA and the generative dynamics of dLLMs. Figure 1 clearly illustrates standard LoRA's limited loss reduction at high noise levels, providing strong intuitive evidence that static weights fail to adapt to the dynamically shifting diffusion process.

- Solid Theoretical Foundation: Theorem 4.1 provides a constructive proof guaranteeing that NaRA's decomposed structure possesses sufficient expressive capacity to equivalently represent a set of independent LoRA matrices trained for discrete timesteps.

Weaknesses:

- Marginal Performance Gains on Certain Tasks. While NaRA achieves SOTA across the board, the margin of improvement over standard LoRA is somewhat incremental in specific settings. For instance, on mathematical reasoning tasks with the LLaDA-8B-Instruct backbone, NaRA achieves an average accuracy of 80.89%, while standard LoRA reaches 80.07% (a mere 0.82% absolute improvement). This raises questions about the practical necessity of the proposed method when fine-tuning heavily aligned/instruction-tuned base models.

- About Block-wise Early Termination. In Section 4.4 and Appendix C, the authors introduce a block-wise early termination mechanism to accelerate inference. While this is a neat engineering optimization that reduces total inference time by ~81%, it is not a feature unique to NaRA (as shown in Table 7, standard LoRA benefits equally from this mechanism). Featuring this prominently in the main text distracts from the paper's core scientific contribution (the noise-aware adaptation mechanism).

---

> ### Author Rebuttal · Authors · 2026-03-31
>
> We sincerely thank the reviewer for recognizing the solid theoretical foundation of our work and the compelling motivation behind NaRA. We address each concern below.
>
> ---
>
> **Weakness 1**: Marginal Performance in Some Tasks
>
> **Answer**: We thank the reviewer for this insightful comment. We agree that on some individual benchmarks, such as mathematical reasoning with the LLaDA-8B-Instruct backbone, the gain over standard LoRA is modest (e.g., +0.82% on average accuracy). However, we believe this should be interpreted in the context of PEFT, where the trainable capacity is intentionally constrained and strong instruction-tuned backbones already provide a high performance ceiling.
>
> Importantly, NaRA consistently outperforms standard LoRA on 12 out of 14 benchmarks across commonsense reasoning, mathematical reasoning, and code generation, demonstrating that the improvement is broad and stable rather than limited to a few tasks. In particular, NaRA brings a notable average gain of 3.91% on commonsense reasoning, and also improves mathematical and code generation performance, e.g., +2.82% on AQuA (57.27% vs. 54.45%) and +1.40% on MBPP (38.60% vs. 37.20%).
>
> We would also like to emphasize that NaRA uses the same training pipeline and supervision as standard LoRA, without introducing extra annotations or changing the optimization objective. This suggests that the observed gains come from the proposed architecture itself, which makes NaRA a lightweight yet effective PEFT method. In heavily aligned/instruction-tuned settings, even moderate but consistent improvements are practically meaningful, especially when they are achieved without additional training cost or supervision.
>
> Overall, we believe NaRA provides a general and reliable improvement over LoRA across diverse tasks, including settings where the backbone is already strong.
>
> ---
>
> **Weakness 2**: About Block-wise Early Termination
>
> **Answer**: We agree that block-wise early termination is a general decoding optimization and should not be interpreted as a NaRA-specific advantage. Standard LoRA also benefits from it. We will revise the paper to make this distinction explicit and remove any wording that could suggest that NaRA itself is inherently faster than LoRA.
>
> ---
>
> **Question 1**: Comparison with Time-interval LoRA Baseline
>
> **Answer**: We thank the reviewer for the helpful suggestion. We implemented a Multi-LoRA baseline that splits the noise range into 4 discrete intervals and uses a separate LoRA adapter for each interval. All other settings are kept the same, and we report the mean and standard deviation over three seeds.
>
> |Method| Params (%) | GSM8K | AddSub | AQuA | MultiArith | Avg |
> |-|-|-|-|-|-|-|
> |Multi-LoRA| 1.68% | 77.18 ± 1.07 | 87.51 ± 0.52 | 37.79 ± 1.38 | 96.83 ± 0.70 | 74.83 |
> |LoRA*| 0.42% | 78.96 ± 0.42 | 88.38 ± 0.87 | 54.45 ± 1.90 | 98.50 ± 0.29 | 80.07 |
> |NaRA*| 0.43% | **79.03 ± 0.24** | **88.53 ± 0.52** | **57.27 ± 0.20** | **98.72 ± 0.63** | **80.89** |
>
> Note: Results marked with (\*) are sourced from our original submission.
>
> As shown in the table, Multi-LoRA performs worse than both LoRA and NaRA despite using about 4× more parameters. This supports our design motivation in the Introduction: "By modeling the core matrix as a continuous function, NaRA naturally captures dependencies across noise levels through the diffusion process in dLLMs without training disjoint adapters." In contrast, time-interval adapters split the training data and prevent information sharing across noise levels, while NaRA enables continuous cross-noise-level learning with parameter efficiency.
>
> ---
>
> **Question 2**: Peak VRAM Comparison
>
> **Answer**: We appreciate this question. For a fair comparison, we re-ran LoRA and NaRA on the same GPU (NVIDIA GeForce RTX 3090) for the code generation task, and report the mean and standard deviation over three seeds below.
>
> | Method | Train Time (min) |Inference Time (sec/sample)| Peak VRAM (GB) |
> |-|-|-|-|
> |LoRA (r=32)| 326.6 ± 15.2 | 47.63 ± 4.66 | 20.18 |
> |NaRA (r=32)| 316.9 ± 28.8 | 53.02 ± 6.95 | 20.19 |
> |**Overhead**| -3.0% | +11.3% | +0.05% |
>
> According to the results, NaRA adds only a negligible VRAM overhead, since its globally shared hypernetwork produces small $r \times r$ matrices. Training time is comparable to LoRA, and the inference overhead is modest and acceptable for practical deployment.
>
> ---
>
> We appreciate the reviewer's constructive feedback and hope these responses address the concerns. We are happy to provide additional experiments or clarifications as needed.

---

> > ### Author Rebuttal · Reviewer_j83D · 2026-04-03
> >
> > Thanks to the authors for their response. I have updated my score accordingly.

---

> > > ### Author Response · Authors · 2026-04-03
> > >
> > > Dear Reviewer j83D,
> > >
> > > Thank you very much for your helpful suggestion, as well as the time and effort you devoted to reviewing our work.
> > >
> > > Sincerely,
> > >
> > > The authors of Submission 15639

---

### Official Review · Reviewer_224S · 2026-03-11

**Soundness:** 2
**Presentation:** 3
**Significance:** 2
**Originality:** 3
**Overall Recommendation:** 3
**Confidence:** 4

**Summary:**

This paper presents NaRA, a LoRA variant that basically uses different LoRA adapters for different stages of the denoising process of the discrete diffusion language model. This is done by using a hypernetwork C($\lambda$) that conditions on the current stage of generation $\lambda$). The hypernetwork then produces the weight update as $A$ $C$($\lambda$) $B$.
The paper provides results on commonsense reasoning and mathematical reasoning tasks which is gold standard in PEFT literature.

**Compliance With Llm Reviewing Policy:**

Affirmed.

**Final Justification:**

I will maintain my score.

It appears as if NaRA's gains diminish with multi-token decoding. A more thorough stress-tested analysis on diffusion-aligned multi-token setting would strengthen the claims.

**Key Questions For Authors:**

Questions:

1. Could you provide results with a constant hypernetwork (not conditioned on time)? It would be helpful to see this on both Commonsense Reasoning and Math tasks.
2. How are the commonsense reasoning tasks formulated? Is it a generative task or a classification/mcq style task where the answer with the lowest perplexity is selected?
3. How is the learning rate selected? I checked the appendix but couldn’t find a specific search strategy. Recent work: https://arxiv.org/abs/2602.04998 provides some evidence that different LRs are optimal for different LoRA variants and when that is controlled for, all LoRA variants converge to similar performance.

**Limitations:**

N/A

[1] It would be useful to discuss whether NaRA can be applied to single-token tasks like classification, or perplexity based evaluations because there is no concept of time conditioning there.

**Strengths And Weaknesses:**

Soundness:

1. The setup behind how hypernetwork modulates the noise is well formulated.
2. It is not clear to me why this is a motivation for diffusion in particular. I believe that the gain is coming from having a dynamic modulator that can in principle be applied over the course of any kind of generative process, not necessarily diffusion.  I understand the attempt at a connection to different noising levels but that is not unique. For example, you can also formulate it as hypernetwork C($\lambda$) where $\lambda$ is the proportion of tokens generated in AR models. So the premise of NaRA that it needs to be specifically coupled with discrete diffusion seems a little shaky to me. This is my main concern with the story as currently presented in this paper.
3. It would be more comprehensive if comparison is provided against various other kinds of PEFT methods such as using singular values [1,2]. If NaRA can demonstrably be applied on top of these methods too instead of just LoRA, as the main idea is the time-conditioned hypernetwork that produces scaling modifications to the core PEFT method, that's a plus.

Significance:

1. This work takes a step towards bridging a timely concept (diffusion language models) with parameter efficient fine-tuning. It is definitely interesting to ask whether certain kinds of PEFT methods are better for diffusion as compared to AR. However, I am less confident as to whether the paper, as currently framed, is the right way to approach this question as it is not diffusion-specific (see point 2 of soundness). A study that explains why NaRA, or related works are more suited to diffusion over AR would immensely strengthen this.

Presentation:

1. Overall the paper is presented quite well. The method is clear and concise, figures and tables are easy to understand. Some experimental details could be more thorough (see Questions).
2. Multiple domains spanning commonsense reasoning, mathematical reasoning, and code-generation are studied.


Originality:

1. I have seen variants of “dynamic LoRA” before (AdaLoRA), including using hypernetworks (Doc-to-LoRA, Text-to-LoRA, HyperAdaLoRA). But this specific variation of hypernetwork conditioned on time producing an update modification is new that I do not recall seeing before.


[1]  https://arxiv.org/abs/2404.02948

[2] https://arxiv.org/abs/2405.19597

---

> ### Author Rebuttal · Authors · 2026-03-31
>
> We thank the reviewer for constructive comments.
>
> ---
>
> **Answer to Soundness & Significance:**
> We thank the reviewer for raising this insightful concern. We agree that the core idea of NaRA is not mathematically restricted to diffusion models, and in principle, a dynamic adapter could also be applied to AR generation. Since dLLMs induce a much more diverse input distribution, NaRA makes noise-level-specific parameterization more necessary.
>
> Specifically, let $U = \{1,\dots,L\}$ denote the full set of token positions. At noise level $\lambda$, we select a subset $S \subseteq U$ of size $m=\lfloor L(1-\lambda)\rfloor$ as the revealed positions, and let $M = U \setminus S$ denote the masked positions. We use $\mathbf{x}_S$ and $\mathbf{x}_M$ to denote the tokens at the revealed and masked positions, respectively.
>
> - For **AR models**, the revealed positions are always the prefix $\{1,\dots,m\}$, and the model predicts $p(x_{m+1}\mid \mathbf{x}_{1:m})$.
>
> - For **dLLMs**, the model conditions on the partially masked input $\mathbf{x}_S$ and predicts the masked tokens $p(\mathbf{x}_M \mid \mathbf{x}_S)$. Since there are $\binom{L}{m}$ possible choices of $S$, the number of possible input configurations for dLLMs is exponentially larger than for AR models.
>
> This combinatorial explosion makes dLLMs a more challenging modeling problem at each noise level than AR models, which is precisely why we think noise-level-specific parameterization is more important for dLLMs. This intuition is also consistent with our response to Reviewer j83D (Question 1): simply partitioning the denoising process into discrete intervals and training separate adapters (Multi-LoRA) performs worse than NaRA, suggesting that effective adaptation in dLLMs requires not only noise awareness, but also continuous sharing across noise levels.
>
> As suggested by the reviewer, we further evaluate whether NaRA can be applied beyond LoRA. In particular, we instantiate NaRA on top of DoRA [1] by introducing a λ-conditioned scaling matrix. The results on code generation tasks are summarized in the table below.
> |Method|Params|HumanEval|MBPP|Avg|
> |-|-|-|-|-|
> |DoRA|0.423|39.02|36.70|37.86|
> |DoRA+NaRA|0.432|**39.84**|**38.47**|**39.16**|
>
> As shown in the table, NaRA's plug-and-play design remains effective when combined with DoRA, leading to improved dLLM performance.
>
> [1] DoRA, ICML 2024
>
> ---
>
> **Answer to Q1:** We conducted ablation experiments with NaRA-C, a variant in which the matrix $C$ in NaRA is made learnable but no longer conditioned on the noise level. All other settings are kept identical to those in the NaRA experiments. The corresponding LoRA and NaRA results are shown in Tables 1 and 2 of the main paper.
>
> |Method|GSM8K|AddSub|AQuA|MultiArith|Avg|
> |-|-|-|-|-|-|
> |NaRA-C|78.43|89.12|55.12|98.50|80.29|
>
> |Method|ARC-C|ARC-E|BOOLQ|HELLA|OBQA|PIQA|SIQA|WINO|Avg|
> |-|-|-|-|-|-|-|-|-|-|
> |NaRA-C|79.92|88.62|55.61|75.98|74.00|83.64|63.13|24.13|68.13|
>
> According to the results, we can see that NaRA-C is slightly better than LoRA on math tasks but still worse than NaRA, and it underperforms LoRA and NaRA on commonsense tasks. This suggests that a noise-independent $C$ matrix is insufficient, supporting the need for noise-level-specific parameterization.
>
> ---
>
> **Answer to Q2:** Our commonsense reasoning evaluation uses a generative formulation by following [2,3]. Given a question, the model generates an answer, from which we extract the predicted answer and compute accuracy by comparing it with the ground truth.
>
> [2] HiRA, ICLR 2025
> [3] P-tuning, ACL 2022
>
> ---
>
> **Answer to Q3:** We selected the best learning rate for each method on a held-out subset of the validation set for the math tasks. The table below reports the corresponding test-set results, which are consistent with these LR choices and suggest that the observed improvement is unlikely to be explained by a mismatched optimal learning rate.
>
> |Method|LR|GSM8K|AddSub|AQuA|MultiArith|MathAvg|
> |-|-|-|-|-|-|-|
> |LoRA|5e-5|74.83|86.67|49.61|93.00|76.03|
> |LoRA|1e-4*|78.96|88.38|54.45|98.50|80.07|
> |LoRA|2e-4|72.91|88.78|50.79|95.11|76.90|
> |NaRA|5e-5|74.93|86.24|49.08|91.94|75.55|
> |NaRA|1e-4*|79.03|88.53|57.27|98.72|80.89|
> |NaRA|2e-4|73.47|89.87|48.56|95.17|76.77|
>
> \* denotes the learning rate used in the main experiments.
>
> ---
>
> **Answer to Limitations:** NaRA is mainly designed for multi-step denoising, where different noise levels appear across inference steps and noise-aware adaptation is useful. For single-token tasks such as classification or perplexity-based evaluation, the inference stage is fixed, so the benefit of noise-aware adaptation is much less pronounced. NaRA can still be applied in principle, but we expect its advantage to be limited in such settings.
>
> ---
>
> We hope these responses address the reviewer's concerns. We are happy to provide additional experiments or clarifications as needed.

---

> > ### Author Rebuttal · Reviewer_224S · 2026-04-01
> >
> > I thank the authors for their rebuttal which addresses many of my questions.
> >
> > Regarding the answer to limitations, the authors state that “NaRA is mainly designed for multi-step denoising.” I would appreciate further clarification on this point. As per Appendix G, it appears that inference is performed one token at a time. However, dLLMs are primarily motivated with the potential to operate in a multi-token denoising regime.
> >
> > Could the authors comment on NaRA’s behavior in this setting? In particular, do they have any experimental results or qualitative observations on different levels of multi-token denoising?

---

> > > ### Author Response · Authors · 2026-04-03
> > >
> > > We thank the reviewer for the follow-up comment.
> > >
> > > We would like to clarify the design choice regarding single-token decoding. Following the experimental setup in the original LLaDA paper, specifically Appendix B.4 of [LLaDA, arXiv:2502.09992](https://arxiv.org/pdf/2502.09992), we adopt one-token-per-step decoding to align with their established evaluation protocol. In our view, the key reason for this choice is that the LLaDA-8B-Instruct model itself exhibits performance degradation when decoding multiple tokens per step, so single-token decoding helps preserve generation quality.
> > >
> > > Importantly, this does not conflict with NaRA's design. NaRA and dLLMs share the same denoising objective, but NaRA uses different adapter parameterizations for different noise levels to better capture the behavior at each denoising stage. This design is compatible with different decoding granularities, regardless of whether one token or multiple tokens are decoded per step, since the noise level still characterizes the current denoising state.
> > >
> > > We also conduct experiments on the code generation task to examine NaRA under multi-token decoding, fixing the generation length at 512 and using 256 or 512 decoding steps.
> > >
> > >
> > > | Method | Steps | Length | HumanEval | MBPP | Avg |
> > > |--------|-------|--------|-----------|------|------|
> > > | LoRA   | 256   | 512    | 27.75     | 30.50 | 29.13  |
> > > | LoRA   | 512   | 512    | 39.43     | 37.20 | 38.32  |
> > > | NaRA   | 256   | 512    | 29.88     | 29.90 | 29.89  |
> > > | NaRA   | 512   | 512    | 40.04     | 38.60 | 39.32  |
> > >
> > > As shown in the table, NaRA consistently outperforms LoRA under both decoding settings. When decoding two tokens per step, NaRA still maintains its advantage over LoRA (29.89 vs. 29.13). This demonstrates that NaRA's noise-aware adaptation is effective beyond single-token decoding.
> > >
> > > We hope this clarifies the reviewer's concern. We sincerely thank the reviewer for the thoughtful question and careful reading of our paper.

---

### Official Review · Reviewer_e3nF · 2026-03-13

**Soundness:** 3
**Presentation:** 3
**Significance:** 3
**Originality:** 3
**Overall Recommendation:** 4
**Confidence:** 4

**Summary:**

The paper introduces Noise-aware Low-Rank Adaptation (NaRA), a Parameter-Efficient Fine-Tuning (PEFT) framework specifically designed for Diffusion Large Language Models (dLLMs). The authors identify a "structural misalignment" in standard LoRA: it applies static updates across the entire denoising trajectory, ignoring that dLLMs face shifting input distributions and generation difficulties as noise levels ($\lambda$) change.
NaRA addresses this by inserting a dynamic core matrix $C(\lambda)$ between two static LoRA matrices. This core matrix is generated by a lightweight, globally shared hypernetwork conditioned on the noise level via Gaussian Fourier embeddings. The authors provide theoretical proof that NaRA can represent multiple independent LoRA adapters across different noise levels and demonstrate its effectiveness on the LLaDA model family across commonsense reasoning, math, and code generation tasks.

**Compliance With Llm Reviewing Policy:**

Affirmed.

**Final Justification:**

The authors have addressed my concerns. I retain my weak acceptance due to the marginal performance improvement from the proposed method.

**Key Questions For Authors:**

Can this approach also benefit other diffusion models in other domains apart from NLP (i.e. masked diffusion models)? Such evidence will broaden the impact of this work.

**Limitations:**

yes

**Strengths And Weaknesses:**

Strengths

The introduction of a hypernetwork to modulate a core matrix is a parameter-efficient way to achieve continuous adaptation without training disjoint adapters for different timesteps.
The design is meant for "plug-and-play" use. By sharing a single hypernetwork across all layers and modules, NaRA adds a small number of additional trainable parameters compared to standard LoRA.


Weaknesses

Marginal Performance in Some Tasks: While NaRA leads on average, the margin over standard LoRA is relatively thin or worse in certain benchmarks.
Comparison Fairness: While the authors mention controlling parameter counts, the total trainable parameters for NaRA are slightly higher than LoRA due to the hypernetwork. A comparison against a LoRA baseline with a slightly higher rank ($r$) to match the exact parameter count would be more definitive.

---

> ### Author Rebuttal · Authors · 2026-03-31
>
> We sincerely thank the reviewer for recognizing the novelty of our hypernetwork-based modulation design and its parameter efficiency. We address each concern below.
>
> **Question1**: Marginal Performance in Some Tasks
>
> **Answer**: We appreciate the reviewer’s observation. Indeed, the performance gains of NaRA vary across benchmarks, with larger improvements on some tasks and smaller margins on others. However, NaRA still improves over the standard LoRA baseline on 12 out of 14 benchmarks across commonsense reasoning, mathematical reasoning, and code generation tasks. In particular, NaRA achieves an average accuracy increase of 3.91% on commonsense reasoning benchmarks. For mathematical and code generation tasks, NaRA improves accuracy by +2.82% on AQuA (57.27% vs 54.45%) and +1.40% on MBPP (38.60% vs 37.20%). These results indicate that NaRA provides broad performance gains rather than improvements confined to a few tasks.
>
> To isolate the effect of the proposed architecture, we keep the training pipeline identical to that used for standard LoRA, despite the additional optimization complexity introduced by the hypernetwork [1,2]. As an early study on PEFT for dLLMs, we believe that the current results already demonstrate the benefit of noise-aware dynamic adaptation under a standard training recipe.
>
> **Question2**: Comparison Fairness
>
> **Answer**: We appreciate the reviewer’s suggestion and further evaluate LoRA with a higher rank, r=40, which has a slightly larger parameter budget than NaRA (r=32) (0.52% vs. 0.43%). Apart from the rank, all other settings are kept identical to the LoRA (r=32) experiments. We report the mean and standard deviation over three random seeds for all experiments.
>
> | Method | Rank | Params (%) | GSM8K | AddSub | AQuA | MultiArith | Avg |
> |--------|-----|------------|-------|--------|------|------------|----------|
> | LoRA | 40 |0.52% | **79.08 ± 0.74** | 87.17 ± 0.73 | 55.91 ± 2.23 | 98.61 ± 0.29 | 80.19 |
> | LoRA* | 32 |0.42% | 78.96 ± 0.42 | 88.38 ± 0.87 | 54.45 ± 1.90 | 98.50 ± 0.29 | 80.07 |
> | NaRA* | 32 |0.43% | 79.03 ± 0.24 | **88.53 ± 0.52** | **57.27 ± 0.20** | **98.72 ± 0.63** | **80.89** |
>
> Note: Results marked with (\*) are sourced from our original submission.
>
> On these mathematical benchmarks, LoRA (r=40) improves over LoRA (r=32) but still underperforms NaRA (r=32), indicating that increasing rank alone does not close the gap.
>
>
> **Question3**: Application Beyond NLP
>
> **Answer**: We appreciate this constructive suggestion. NaRA is designed to modulate LoRA adapters via a noise-conditioned hypernetwork, and this mechanism is modality-agnostic in principle. In other words, it is not specific to the NLP setting alone; rather, it can be applied to other diffusion-based generative models as long as they expose a timestep/noise-conditioning signal.
>
> In this work, we focus on dLLMs to systematically evaluate NaRA on diverse text-based reasoning tasks, including commonsense reasoning, mathematical reasoning, and code generation. We agree that extending NaRA to other modalities, such as image or video generation, is an interesting future direction, and we will discuss this broader applicability in the revision.
>
> [1] Charakorn et al., "Text-to-LoRA: Instant Transformer Adaption," ICML 2025.
>
> [2] Lv et al., "HyperLoRA: Efficient Cross-task Generalization via Constrained Low-Rank Adapters Generation," EMNLP 2024.
>
> ---

---

> > ### Author Rebuttal · Reviewer_e3nF · 2026-04-02
> >
> > The authors have addressed my concerns. I retain my weak acceptance due to the marginal performance improvement from the proposed method.

---

> > > ### Author Response · Authors · 2026-04-03
> > >
> > > Dear Reviewer e3nF,
> > >
> > > Thank you again for your constructive feedback. We would like to bring one additional update to your attention, directly related to your key question regarding applicability beyond NLP.
> > >
> > > In our initial response, we noted that NaRA is modality-agnostic in principle, but at that time we could only promise to discussing broader applicability in the revision. Since then, we have conducted an additional experiment in the image diffusion domain to directly address this concern. Specifically, we applied NaRA to SDXL under the DreamBooth subject-driven generation setup [1]. Following the official HuggingFace `Diffusers` example, we use the `google/dreambooth` dataset, focusing on the `dog6` subset with 5 images, and train for 1000 steps. During training, NaRA takes the normalized diffusion timestep $\lambda = t/T \in [0,1]$ as the noise-level input to the hypernetwork. All other training settings are kept the same as standard LoRA.
> > >
> > > For evaluation, following [1], we generate 8 images per method using DDIM with the prompt "a photo of sks dog", and evaluate them using DINO, CLIP-I, and FID. The results are shown below.
> > >
> > > | Method | DINO ↑ | CLIP-I ↑ | FID ↓ |
> > > |-|:-:|:-:|:-:|
> > > | LoRA | 0.0458 | 0.6901 | 22.13 |
> > > | **NaRA** | **0.0610** | **0.7365** | **16.71** |
> > >
> > > These results show that NaRA outperforms LoRA on all three metrics, demonstrating that noise-aware PEFT is also effective in the image diffusion domain. We hope this additional experiment addresses your original concern about the broader impact of our method. We sincerely thank you for your thoughtful suggestion and the time you devoted to evaluating our work.
> > >
> > > [1] DreamBooth, CVPR 2023
> > >
> > > Sincerely,
> > >
> > > The authors of submission 15639

---

### Decision · Program_Chairs · 2026-04-30

**Decision:**

Accept (regular)

**Comment:**

This paper studies a timely and underexplored problem of parameter-efficient fine-tuning for diffusion LLMs. The proposed method, NaRA, is conceptually simple and technically sound. It augments LoRA with a lightweight noise-conditioned hypernetwork so that the low-rank update can adapt across denoising steps. The paper is generally well-written, the method is clearly presented, and the empirical study is reasonably broad, covering commonsense reasoning, mathematical reasoning, and code generation.

The main weakness revealed in reviews is that the empirical gains over strong baselines are mostly modest, and the evaluation remains centered on the LLaDA family. I also share the reviewers’ view that the diffusion-specific motivation could be articulated more sharply, and that the method appears somewhat sensitive to tuning choices. In addition, the timestep-dependent adapter is less deployment-friendly than standard LoRA because it is not as easily merged into the base model.

That said, the rebuttal addressed a substantial portion of the concerns in a meaningful way. In particular, the authors added stronger fairness checks against higher-rank LoRA, a discrete Multi-LoRA baseline, a constant-matrix ablation, additional multi-token decoding results, and clarified the efficiency claims with matched-hardware comparisons. These additions demonstrated that the proposed noise-aware design provides a consistent, lightweight improvement.

Overall, I therefore recommend acceptance, while encouraging the authors to moderate their claims further and more explicitly discuss the remaining limitations in the final version.